# Retention of paternal DNA methylome in the developing zebrafish germline

Ksenia Skvortsova[1,10], Katsiaryna Tarbashevich[2,10], Martin Stehling[3], Ryan Lister [4,5], Manuel Irimia [6,7,8], Erez Raz[2] & Ozren Bogdanovic[1,9]

Two waves of DNA methylation reprogramming occur during mammalian embryogenesis; during preimplantation development and during primordial germ cell (PGC) formation. However, it is currently unclear how evolutionarily conserved these processes are. Here we characterise the DNA methylomes of zebrafish PGCs at four developmental stages and identify retention of paternal epigenetic memory, in stark contrast to the findings in mammals. Gene expression profiling of zebrafish PGCs at the same developmental stages revealed that the embryonic germline is defined by a small number of markers that display strong developmental stage-specificity and that are independent of DNA methylation-mediated regulation. We identified promoters that are specifically targeted by DNA methylation in somatic and germline tissues during vertebrate embryogenesis and that are frequently misregulated in human cancers. Together, these detailed methylome and transcriptome maps of the zebrafish germline provide insight into vertebrate DNA methylation reprogramming and enhance our understanding of the relationships between germline fate acquisition and oncogenesis.

[1] Genomics and Epigenetics Division, Garvan Institute of Medical Research, Sydney, NSW 2010, Australia. [2] Institute of Cell Biology, Center for Molecular Biology of Inflammation, University of Münster, Münster 48149, Germany. [3] Flow Cytometry Unit, Max-Planck-Institute for Molecular Biomedicine, Roentgenstraße 20, 48149 Münster, Germany. [4] ARC CoE Plant Energy Biology, School of Molecular Sciences, The University of Western Australia, Perth, WA 6009, Australia. [5] Molecular Medicine Division, Harry Perkins Institute of Medical Research, Perth, WA 6009, Australia. [6] Centre for Genomic Regulation, The Barcelona Institute for Science and Technology, Barcelona 08003, Spain. [7] Universitat Pompeu Fabra (UPF), Barcelona 08002, Spain. [8] ICREA, Barcelona 08010, Spain. [9] School of Biotechnology and Biomolecular Sciences, University of New South Wales, Sydney, NSW 2010, Australia. [10] These authors contributed equally: Ksenia Skvortsova, Katsiaryna Tarbashevich. Correspondence and requests for materials should be addressed to O.B. (email: o.bogdanovic@garvan.org.au)

Vertebrate embryogenesis requires tight orchestration of spatiotemporal gene expression patterns. This is achieved through the coordinated action of transcription factors, diverse signalling molecules and genomic regulatory marks such as DNA methylation (5-methylcytosine; 5mC) and histone tail modifications[1]. The acquisition of these regulatory determinants defines key developmental stages and is implicated in processes such as pluripotency establishment and cell differentiation[2,3]. 5mC is a chemical modification of the DNA, mostly associated with transcriptional repression and highly abundant on CpG dinucleotides in vertebrate genomes[4,5]. The addition of 5mC to genomic DNA is catalysed by DNA methyltransferases (DNMTs)[6,7], whereas its removal can occur via active and passive DNA demethylation mechanisms. An active mechanism entails enzymatic oxidation of genomic 5mC by TET (Ten-eleven-translocation) family enzymes[8], whereas the passive mechanism is primarily linked to the temporary exclusion of the maintenance methyltransferase (DNMT1) activity from the nucleus[9–11].

During mammalian preimplantation development and primordial germ cell (PGC) formation, the DNA methylome is erased and then re-established[12,13]. In mammalian zygotes, the paternal genome is rapidly demethylated shortly after fertilisation, followed by a progressive drop in 5mC of both paternal and maternal genomic contributions. DNA demethylation takes place up until the blastocyst stage followed by cell-type-specific remethylation during gastrulation[13]. While the exact mechanism by which DNA demethylation occurs in the mammalian zygote remains a topic of debate[14–17], recent data suggest that TET proteins are not directly implicated in the initial wave of paternal DNA demethylation, which due to its dynamics can also not be fully explained by passive 5mC dilution[17].

Similarly, during mammalian PGC DNA methylome reprogramming, TET proteins are not required for the initiation, but rather for the maintenance of global PGC demethylation[18]. It has been proposed that one of the major functions of this PGC DNA demethylation event is the erasure and resetting of genomic imprints in the developing germline[12,19]. Importantly, not all genomic sequences are targeted for demethylation with the same efficiency, thereby constituting a potential platform for inter- and transgenerational epigenetic inheritance[20].

Zebrafish and other non-mammalian (anamniote) vertebrates lack global 5mC erasure[21–26], which in mammals occurs after fertilisation and persists during blastula stages[27,28]. Zebrafish instead inherit the paternal DNA methylome configuration[25,26]. During cleavage stages, before the onset of zygotic genome activation (ZGA), the maternal epigenomic contribution is remodelled through the acquirement of 'placeholder' nucleosomes[29]. Placeholders are hypomethylated domains in the genome, which are enriched in the histone modification H3K4me1 and the histone variant H2AZ. Notably, TET activity has not been detected in pluripotent anamniote embryos[26,30–32]. Both anamniotes and mammals, however, employ TET-dependent demethylation for the regulation of enhancer chromatin during the vertebrate phylotypic stage, the most conserved phase of vertebrate embryogenesis, characterised by pan-vertebrate gene-regulatory similarities[32,33]. It is not yet clear how these differences in epigenome remodelling between anamniotes and mammals relate to their diverse developmental strategies, for example, the different timing of ZGA[34] and the lack of intrauterine development in anamniotes.

A major unresolved question in the field of developmental epigenetics is whether the PGC epigenome remodelling event observed in mammals is a conserved feature of vertebrate embryogenesis. To date, detailed epigenome and transcriptome maps have only been generated for the developing mouse and human germline[16,18,35–41]. Notably, germline specification in mammals and urodele amphibians utilises a different strategy

than that of *Drosophila*, *Xenopus*, and zebrafish. In the former group the germline is specified in the early embryo by means of induction, which in mice occurs via BMP/Wnt signalling[42], whereas in the latter group the determinants required for PGC specification are maternally provided[43,44]. These determinants, collectively referred to as germ plasm, contain RNA and proteins corresponding to key germline components. These differences notwithstanding, a small number of germline determinants such as *ddx4 (vasa)*, *dazl*, *dnd* and *nanos* are expressed during both murine and zebrafish PGC development[44]. Migration of germ cells from the site of specification to the position of the gonad development is another feature common in vertebrate and many invertebrate organisms[45]. In fish as well as in mammals PGCs are guided by the chemokine Cxcl12a to reach their target[46]. Interestingly, the same or analogous mechanisms of cell guidance and motility are shared with numerous aggressive cancer cells[47–50].

Here we provide whole-genome bisulfite sequencing (WGBS) methylomes[51,52] and transcriptomes of zebrafish PGCs and somatic cells during four stages of embryogenesis. Our data demonstrate the absence of genome-wide 5mC reprogramming in the developing (4−36 h post fertilisation (hpf)) zebrafish germline, in contrast to the findings in mammals. Furthermore, we characterise the zebrafish PGC transcriptome in detail and identify previously uncharacterised germline transcripts, some of which also display germline-specific expression in mammals. Finally, through further exploration of WGBS data we characterise early embryonic targets of 5mC and provide links between embryonic promoter 5mC and misregulation of RNA expression in human cancers.

## Results

**Absence of genome-wide 5mC reprogramming in zebrafish PGCs.** To examine the DNA methylomes and transcriptomes of zebrafish PGCs, we utilised fluorescence-activated cell sorting (FACS) to separate PGCs from somatic cells at different stages of embryogenesis: 4 hpf (blastula), 7 hpf (gastrula), 24 hpf (pharyngula prim-5), and 36 hpf (pharyngula prim-25). The PGCs were sorted from the *kop-EGFP-F′-nos3-′UTR-cry-DsRed* transgenic line[53–55] (Fig. 1a, Supplementary Fig. 1) and were subjected to WGBS methylome and transcriptome (RNA-sequencing (RNA-seq)) library preparation and sequencing (Supplementary Dataset 1). The purity of the sorted PGC cells was estimated to be >97% (Supplementary Fig. 2). The embryonic stages were chosen according to reciprocal best transcriptome similarity index[56], to match the developmental period of mouse PGC specification and DNA methylome reprogramming[18,36] (Fig. 1b). Specifically, we wanted to capture the developmental period, which in mouse would correspond to the initial specification of PGCs and early demethylation (E6.25–E8.5/E9.5), migration and colonisation of the genital ridge (E8.5/E9.5–E10.5), and global DNA demethylation (E10.5–E12.5/E13.5)[18]. It is worth noting that while significant differences in germline development strategies exist between zebrafish and mammals, in both organisms this period is characterised by PGC migration[42–44]. To further assess the purity level of sorted PGC populations, we examined the expression of known germline markers. Indeed, PGC markers, such as *nanos3*, *dnd1*, *tdrd7a*, *ddx4*, *dazl*[57–61], display strong enrichment in the PGC pool over the somatic pool, ranging from 16- to 80-fold (Fig. 2a). We next explored the dynamics of global DNA methylation in PGCs and corresponding somatic cells throughout early development (Fig. 2b). Assessment of 5mC levels of single CpG sites revealed a progressive decrease of the most highly methylated fraction, which was most notable at the 7−24 hpf transition (Fig. 2b), coincident with the onset of *tet* expression and active enhancer demethylation[32]. This progressive decrease

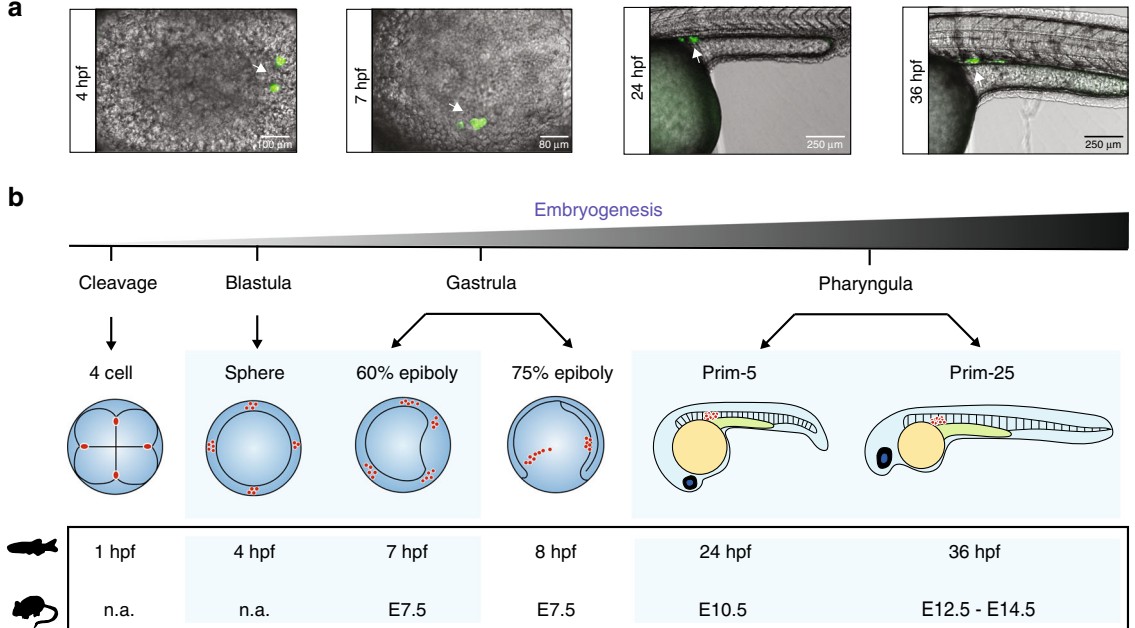

**Fig. 1** Zebrafish germline development. **a** Fluorescence microscopy imaging of zebrafish *kop-EGFP-F'-nos3-'UTR-cry-DsRed* transgenic embryos at 4, 7, 24, and 36 hpf (h post fertilisation). **b** A schematic representation of stages of germline development in zebrafish. At the four-cell stage, germ plasm (red circles) localises to the four cleavage furrows. At 4 hpf (sphere), four clusters of primordial germ cells (PGCs) (red circles) are easily identifiable. In the gastrula stage embryo, four clusters of PGCs start to migrate dorsally. By the pharyngula stage, PGCs have completed their migration and are located between the 8th and the 10th somite

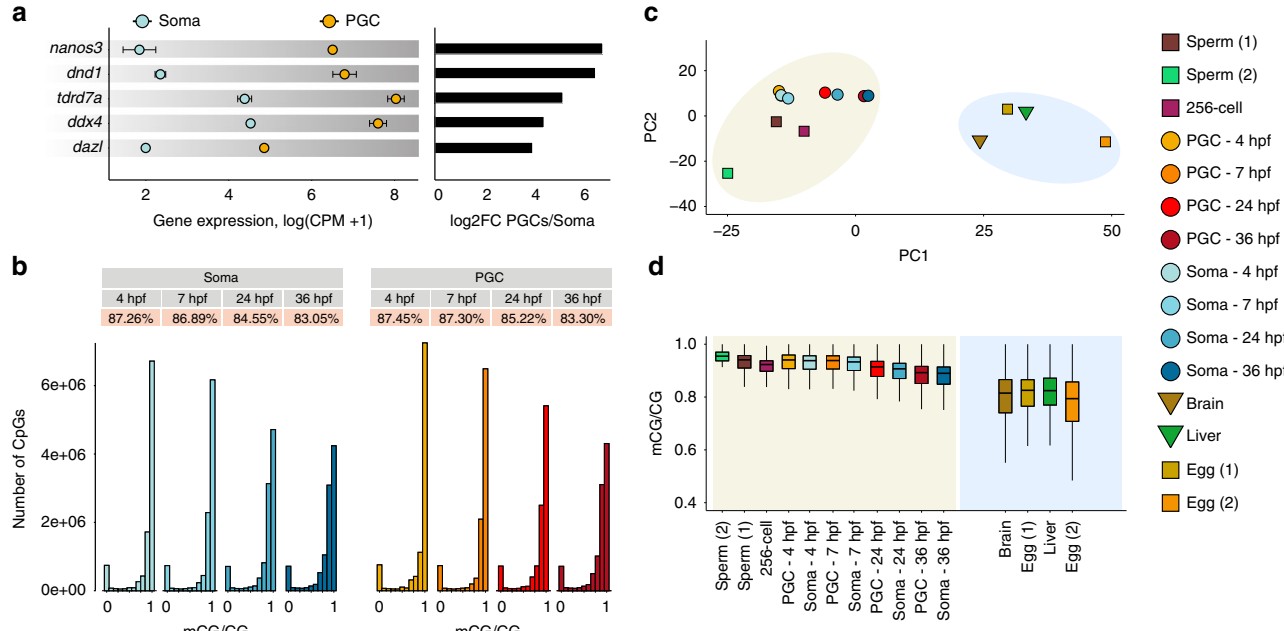

**Fig. 2** Zebrafish retains paternal epigenetic memory in primordial germ cells (PGCs). **a** Gene expression log(CPM + 1) and average enrichment levels of key germline markers in PGCs and somatic cells. Error bars represent standard error (SE), $n = 2$ biologically independent replicates. **b** Histograms of 5-methylcytosine (5mC) levels of single CpG sites in PGCs and somatic cells at four developmental stages. **c** Principal component analysis (PCA) of whole-genome bisulfite sequencing (WGBS) methylomes corresponding to PGCs and somatic cells at 4, 7, 24 and 36 hpf, and sperm, oocyte, liver and brain adult tissues (10 kb non-overlapping genomic bins). **d** Boxplots of genomic 5mC levels (10 kb non-overlapping genomic bins) in embryonic and adult somatic and germ cells. The boxes show the interquartile range (IQR) around the median. The upper and lower whiskers extend from the hinge to the largest and smallest value, respectively, no further than 1.5 IQR

in 5mC content was present in both PGCs and somatic cells, indicative of shared 5mC remodelling mechanisms between the soma and the developing germline (Supplementary Fig. 3a, b). Next, we wanted to test whether genomic 5mC patterns are congruous between PGC and soma samples. Averaged 5mC profiles in non-overlapping 1 kb genomic bins demonstrated strong homogeneity of genome-wide 5mC patterns between the soma and the germline at all developmental stages

examined ($r^2$ (4 hpf) = 0.8, $r^2$ (7 hpf) = 0.89, $r^2$ (24 hpf) = 0.84, $r^2$ (36 hpf) = 0.87; Supplementary Fig. 3c).

Zebrafish genome is characterised by an extremely high repeat content of 52.2%, which is among the highest values reported so far in vertebrates[62]. The majority of sequencing reads that map to repetitive DNA in a WGBS experiment are usually discarded due to mapping to multiple genomic locations. This can potentially lead to inaccurate measurements of global repeat 5mC. To interrogate in detail the repeat 5mC content of somatic cells and PGCs, we generated an in silico reference consisting of canonical repeat sequences ($n = 2322$) from Repbase[63], and the non-repetitive portion of the zebrafish genome. In such a reference, each repeat in its consensus form is represented only once, thus allowing for the mapping of reads that would normally be discarded. Such global assessment of repeat 5mC at the single CpG level did also not reveal any significant differences between PGCs and somatic cells (Supplementary Fig. 4), thus excluding the possibility of global repeat 5mC remodelling in PGCs.

We next performed principal component analysis (PCA) of embryonic PGC, soma and whole embryo methylomes, and compared them to adult germline and somatic methylomes[25,26,32] (Fig. 2c). This unsupervised exploratory analysis identified a notable separation of samples into two well-defined clusters. The first cluster consists of all adult female germline (mature unfertilised oocyte—egg) samples as well as all adult somatic tissues (brain, liver), whereas the second cluster is composed of embryonic soma and PGC methylomes and the adult male germline (sperm). These data demonstrate that the paternal methylome configuration observed in pre-ZGA zebrafish embryos[25,26] is also preserved in the developing germline. It is likely that this clustering pattern is predominantly influenced by global 5mC levels, which are lower in eggs and somatic tissues, and higher in whole embryos, PGCs and sperm (Fig. 2d). It is worth noting, however, that zebrafish oocytes also resemble adult somatic tissues in hypermethylation of germline and developmental promoters, and hypomethylation of housekeeping and terminal differentiation promoters[26]. Overall, this suggests that any potential 5mC remodelling event that is responsible for the remarkable separation of male and female 5mC germline patterns (Fig. 2c, d) is likely to occur later during gonadal development[64]. Zebrafish thus retain paternal epigenetic memory in the germline during PGC migration and likely employ more extensive 5mC remodelling only during later stages of gametogenesis.

**Localised 5mC differences contribute to PGC function.** Having observed absence of global 5mC reprogramming in the early developing zebrafish germline, we next set out to interrogate the DNA methylomes for the existence of locus-specific 5mC differences. We screened the methylome profiles for differentially methylated regions (DMRs, false discovery rate (FDR) <0.05, minimum change in fraction of methylated CpGs ($\Delta$mCG) = 0.1)[65] between age-matched somatic and PGC samples. Furthermore, DMRs overlapping the Repeatmasker track[66], which marks interspersed repeats and low complexity DNA sequences, were removed from further downstream analyses (Supplementary Dataset 2). This analysis resulted in the identification of 157 regions (Fig. 3a, Supplementary Fig. 5a, Supplementary Dataset 3). The majority of these regions were either hypomethylated in 4 hpf PGCs (4 hpf-hypo DMRs, $n = 43$) or hypermethylated in 24 hpf PGCs (24hpf-hyper DMRs, $n = 56$) as compared to their methylation state in somatic cells (Fig. 3b). Both groups of DMRs were mostly located gene distally (Supplementary Fig. 5b). Whereas gene ontology enrichment analysis[67] did not identify any significant associations for the 4 hpf category, the functions of a number of 4 hpf-hypo PGC DMR-linked genes (arpc2, loxl1,

ctnna2, and others) have previously been associated with cell adhesion, cell migration and actin polymerisation[68–70], in line with the migratory nature of PGCs. Other genes such as ccnd2a have previously been described as regulators of meiosis and gametogenesis[71]. These DMRs displayed significant enrichment ($p < 0.001$, Student's $t$ test) in H3K27ac and H3K4me1 histone marks that are associated with active enhancer and promoter chromatin[72,73] (Fig. 3c). Similarly, these DMRs overlapped with CpG islands, which are short genomic sequences of high CpG density that: (i) are usually hypomethylated in vertebrate genomes, (ii) that frequently coincide with gene-regulatory regions and (iii) that can be subject to 5mC-mediated regulation[74,75]. CpG islands were initially defined based solely on their sequence content[76,77]; however, such algorithms did not perform well on anamniote genomes[78]. We thus utilised genomic coordinates of CpG islands identified through the Bio-CAP (non-methylated DNA pulldown) approach throughout this study[75] (Fig. 3c, d).

The major fraction of DMRs ($n = 56$) identified in our analysis were regions that retain 5mC in PGCs at 24 hpf when compared to the soma. As in 4 hpf-hypo DMRs, these loci were significantly enriched ($p < 0.001$, Student's $t$ test) for regulatory DNA elements and active chromatin marks (Fig. 3d). Interestingly, in both PGCs and somatic cells the 24 hpf-hyper DMRs display developmental demethylation, albeit in PGCs this process occurs more slowly (Supplementary Fig. 5c). Moreover, the 24 hpf-hyper DMRs were associated with gene ontology categories implicated in gene regulation and RNA metabolism (Fig. 3e), as well as in reproduction and fertilisation (Supplementary Dataset 4). Genes that contributed to the enrichment of ontologies associated with reproductive processes included dmrt1, lmrp, fzd8a, park7 and szl. Notably, dmrt1 is expressed in zebrafish gonads[79] and linked to male sexual development in zebrafish[80]. We also observed an enrichment ($p = 1.819e − 12$, Wilcoxon's signed-rank test) in 5-hydroxymethylcytosine (5hmC) that was measured in 24 hpf whole embryos[32], over 24 hpf-hyper DMRs, which is indicative of their active, TET protein-mediated demethylation (Fig. 3f). Comparisons with previously published DMRs that lose 5mC developmentally during the phylotypic period and that are associated with CpG islands (phylo-DMRs, $n = 2402$)[32] revealed an overlap of 39% ($n = 22$) (Supplementary Fig. 5d). Indeed this PGC-specific delay in phylo-DMR demethylation was common for the entire phylo-DMR population (Supplementary Fig. 5e). Very few regions were found to specifically retain 5mC in somatic cells (Fig. 3a), supporting the notion that the temporal delay in 5mC removal is a PGC-specific regulatory process. Overall, the genomes of developing PGCs display localised 5mC differences linked to genes implicated in transcriptional regulation and PGC function, and are indicative of differential usage of gene-regulatory regions between the developing germline and the soma.

**Evolutionarily conserved and zebrafish-specific PGC markers.** After characterising the dynamics of 5mC during PGC development, we next wanted to identify genes that display differential expression between PGCs and somatic cells and explore their association with the changes in the epigenome as well as their evolutionary conservation. To thoroughly characterise PGC gene expression programmes, we employed RNA-seq to profile developing germline and soma transcriptomes from age-matched embryos at 4, 7, 24, and 36 hpf (Supplementary Dataset 5). Differential gene expression analysis[81] showed elevated expression levels of the germline markers: nanos3, dnd1, tdrd7a, ddx4, dazl in PGCs at all four examined stages (Supplementary Fig. 6). Genes whose RNAs are highly expressed in PGCs were characterised by gene ontology enrichments[82] associated with

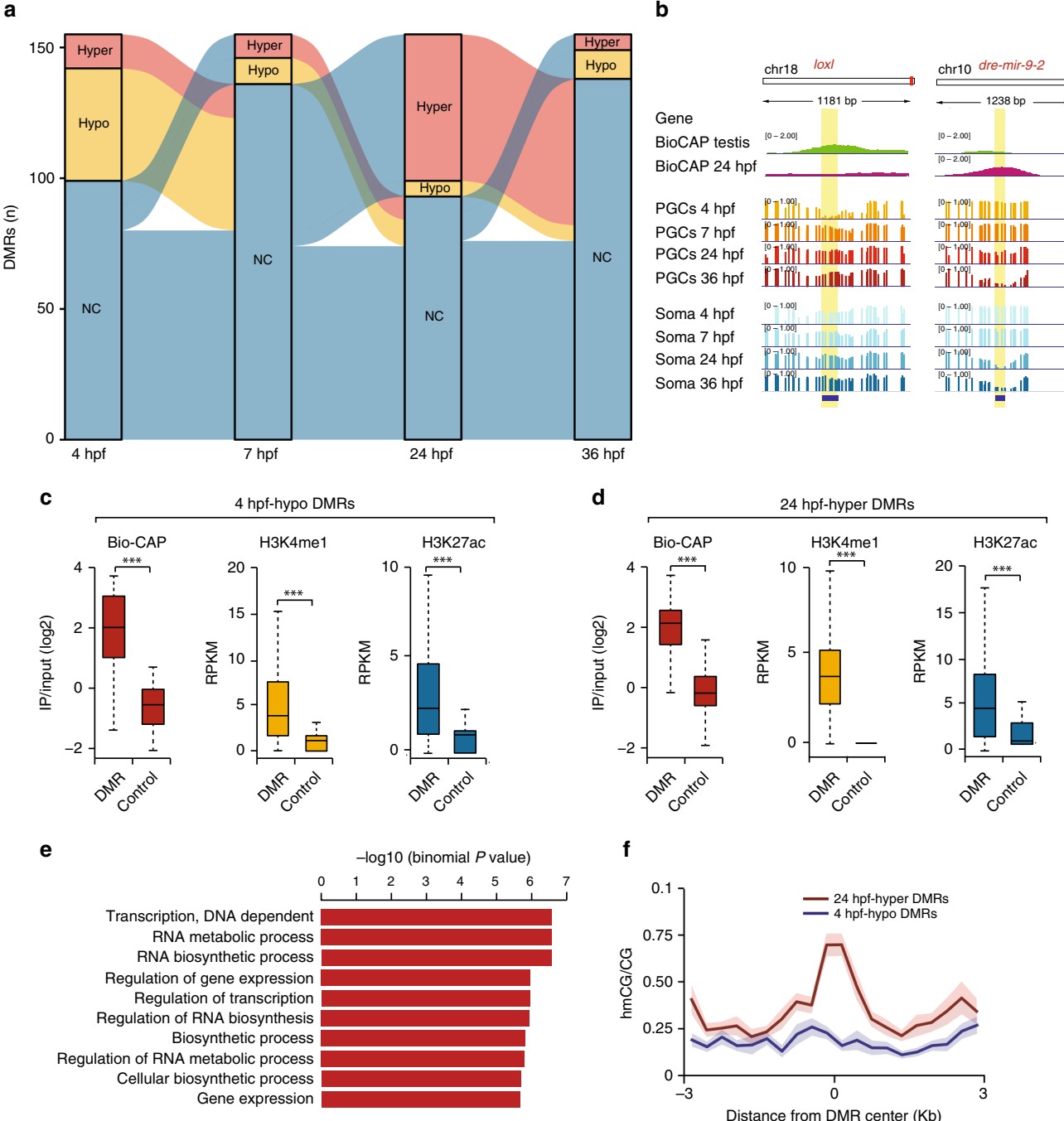

**Fig. 3** Locus-specific 5-methylcytosine (5mC) changes in primordial germ cells (PGCs). **a** Alluvial plot showing the number of hypo- and hypermethylated differentially methylated regions (DMRs) in PGCs compared to somatic cells and their dynamics across four developmental stages. **b** Browser track depicting DNA methylome profiles of 4 hpf-hypo and 24 hpf-hyper DMRs in PGCs and soma at four developmental stages as well as CpG island (Bio-CAP) signal in testis tissue and 24 hpf embryos. **c**, **d** Boxplots showing BioCAP, H3K4me1, and H3K27ac chromatin immunoprecipitation-sequencing (ChIP-seq) signal enrichment at 4 hpf-hypo (**c**) and 24 hpf-hyper (**d**) DMRs compared to a set of size distribution-matched control regions ($p$ value <0.001, Student's $t$ test). The boxes show the interquartile range (IQR) around the median. The upper and lower whiskers extend from the hinge to the largest and smallest value, respectively, no further than 1.5 IQR. **e** Gene ontology (GO) analysis of genes located in the vicinity of 24 hpf-hyper DMRs. **f** Tet-assisted bisulfite sequencing (TAB-seq) (5-hydroxymethylcytosine (5hmC)—from 24 hpf whole embryos) enrichment profiles over 4 hpf-hypo and 24 hpf-hyper DMRs extended 3 kb up- and downstream from the DMR centre, displaying significant ($p$ value = 1.819e − 12, Wilcoxon's signed-rank test) differences. 5hmC values were calculated in 300 bp bins

reproduction, gamete generation and germ-plasm localisation (Supplementary Datasets 6–9). Moreover, we identified 16 genes that were specifically enriched in PGCs throughout the first 36 h of embryonic development (Supplementary Dataset 10). Indeed, these markers were significantly associated with germline-related

ontologies, and included miR-430 RNA binding sites (Fig. 4a), which are frequently found in transcripts that are differentially stabilised between somatic and germ cells[83–85].

We next sought to identify RNAs enriched in PGCs and group them according to their developmental expression patterns. To

formulate a list of putative PGC development regulators (FDR <0.05, min logFC = 1.5), we considered genes that are enriched in PGCs in at least two consecutive developmental stages. This resulted in the identification of 47 genes that were divided in five clusters according to their developmental expression dynamics

(Fig. 4b). We next examined whether any of these loci are characterised by 5mC regulation. Genomic intersections of the identified DMRs (Fig. 3a), with the regulatory domains[67] of putative PGC regulators (Fig. 4) revealed no overlap (Supplementary Dataset 3, Source Data File 4), suggestive of 5mC not

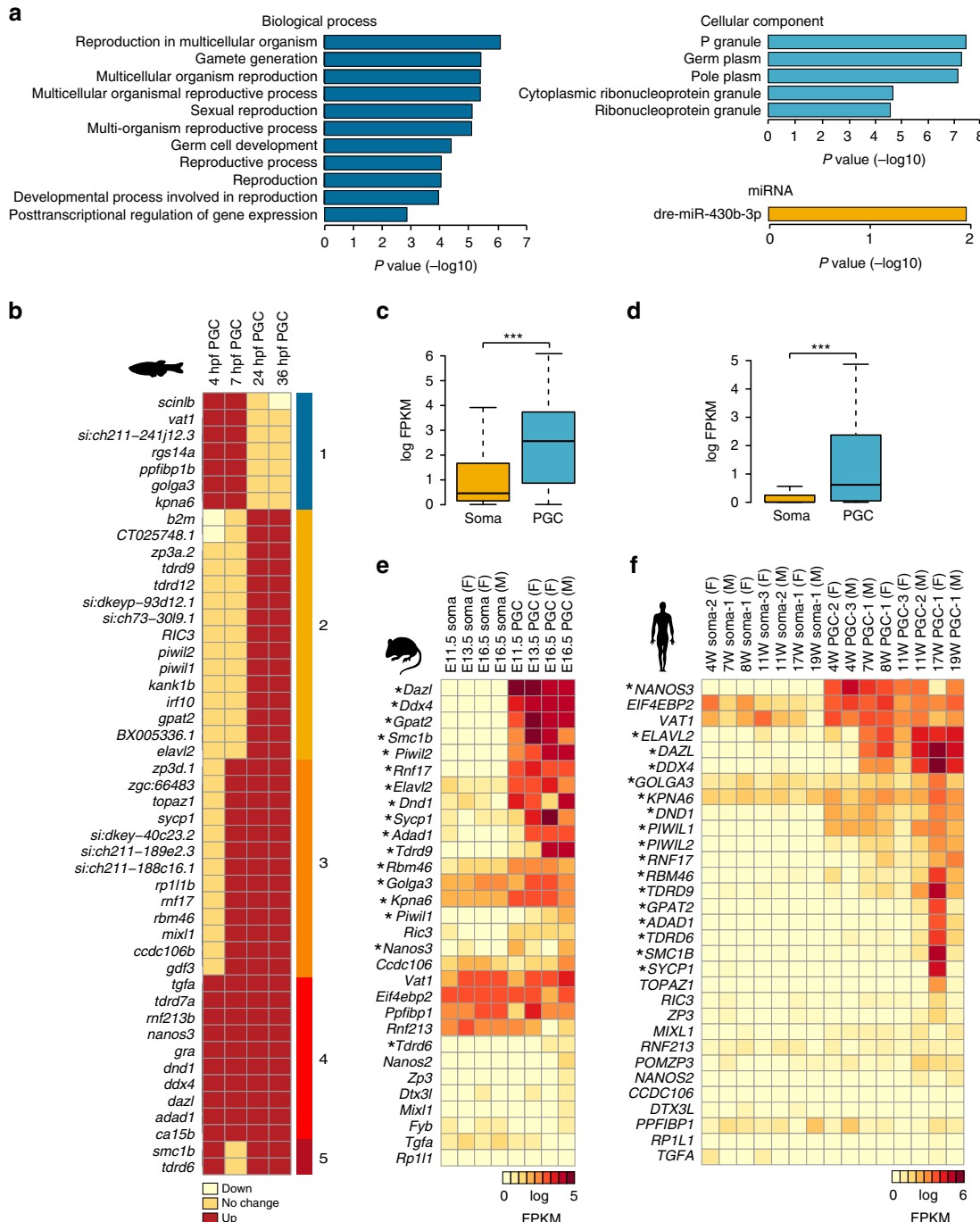

**Fig. 4** Evolutionary conservation of transcriptional programmes in vertebrate primordial germ cells (PGCs). **a** Gene ontology enrichment (Biological Process, Cellular Component, and miRNA binding sites) analysis of genes consistently upregulated in PGCs compared to corresponding somatic cells. **b** Heatmap showing the dynamics of fold changes of genes that display higher transcript abundance in PGCs in at least two consecutive developmental stages: up—upregulated in PGCs (log FC >1.5, false discovery rate (FDR) <0.05), down—downregulated in PGCs (log FC <−1.5, FDR <0.05). **c, d** Boxplots showing log-transformed FPKM (fragments per kilobase million) expression values of the orthologues of putative germline regulators in mouse (**c**) and human (**d**) PGCs and somatic cells (p value <0.001, Student's t test). The boxes show the interquartile range (IQR) around the median. The upper and lower whiskers extend from the hinge to the largest and smallest value, respectively, no further than 1.5 IQR. **e f** Heatmaps showing log-transformed FPKM expression values of putative germline regulators across mouse (**e**) and human (**f**) development, for PGCs and somatic cells. Asterisks indicate genes with evolutionarily conserved PGC-specific expression patterns (conserved in fish, mouse and human)

playing a major role in PGC marker regulation. To investigate whether PGC-specific expression patterns are conserved within the vertebrate subphylum, we identified mouse and human orthologues of the identified putative PGC regulators[86] and assessed their expression in PGCs and matched somatic tissue (Fig. 4c–f)[40,41]. In both mouse (Fig. 4c) and human (Fig. 4d) these RNAs were significantly ($t$ test, $p < 2.2e - 16$) enriched in PGCs. Furthermore, we identified 17 high confidence PGC regulators that displayed PGC-specific expression across all three examined species (Fig. 4e, f). Notably, these genes were strongly associated with ontologies related to PIWI-interacting RNA function (Supplementary Dataset 11). Apart from the well-described germline markers such as *ddx4*, *dazl*, *piwil1* and *piwil2* among others, this analysis also revealed high conservation in the PGC-specific expression of less well-characterised potential germline determinants such as *golga3*, *gpat2*, *kpna6* and *adad1*[87–90].

To take full advantage of the generated PGC transcriptomes, we next sought to explore whether alternative splicing (AS) plays a role in PGC gene regulation. AS has previously been linked to developmental processes and to the establishment and maintenance of cell identity[91–93]. To investigate how AS is regulated between PGCs and soma, we performed deeper coverage RNA-seq for the two replicates at 7 and 24 hpf and quantified alternative exon usage[94]. We compared PGC and soma transcriptomes at each of those time points, identifying 498 exons with different inclusion levels in any pairwise tissue or developmental comparison (Supplementary Fig. 7a). Unsupervised clustering and PCA of inclusion levels of these 498 exons across the eight samples showed a strong separation by developmental stage, with a much smaller tissue contribution to inclusion level variation (Supplementary Fig. 7b, c). Consistently, the majority of differentially regulated exons were identified in pairwise developmental comparisons (7 vs. 24 hpf), with a highly significant overlap between the developmentally regulated exons in PGC and soma (Supplementary Fig. 7a, $p$ value $1.12 \times 10^{-46}$, Fisher's exact test). Interestingly, exons that are differentially utilised between tissues were enriched in gene ontologies associated with regulation of kinase activity, Rho signal transduction and small GTPases, which are essential hallmarks of migratory cells[95] (Supplementary Datasets 12, 13). Lastly, using stringent criteria we could identify seven exons with highly consistent differential usage between PGC and soma across development that can potentially serve as tissue-specific markers (Supplementary Fig. 7d). Altogether, this transcriptome profiling of the developing zebrafish germline provides insights into the developmental dynamics and evolutionary conservation of PGC gene expression and expands the array of zebrafish-specific and conserved PGC markers.

**Somatic and germline targets of promoter 5mC**. Promoters of germline markers such as *ddx4*, *dazl* and *piwil1* have previously been described as targets of embryonic 5mC in zebrafish[26,32]. We therefore wanted to interrogate whether these promoters are targeted by 5mC in an organism-wide fashion or whether this silencing process is confined to somatic but not germline tissues in zebrafish embryos. Our DMR analysis (Fig. 3a) revealed an absence of promoters that undergo soma- or PGC-specific 5mC targeting during embryogenesis. To ensure that such potential targets were not excluded from the analysis due to stringent DMR calling requirements, we identified promoters that overlapped a CpG island[75] and that displayed a developmental 5mC increase in somatic cells starting at 24 hpf. We identified 68 promoters with those features, 65 of which displayed a statistically significant (Fisher's exact test, Benjamini–Hochberg adjusted $p$ value < 0.01)

increase in 5mC at 24 hpf (Supplementary Dataset 14). Importantly, all of these promoters also displayed a developmental increase in 5mC in PGCs (Fig. 5a), in line with the absence of soma- or PGC-specific developmental promoter DMRs. These genes were significantly enriched in ontologies associated with reproductive processes, cell cycle regulation, and protein phosphorylation (Supplementary Dataset 15). Notably, almost all of the identified 5mC target promoters were unmethylated in adult sperm while being methylated in the oocytes, which is in contrast to the predominantly paternal DNA methylome patterns observed during embryogenesis. This list included genes that were previously described as targets of embryonic 5mC in zebrafish, including *ddx4*, *dazl* and *piwil1*[26,32]. However, we also identified previously uncharacterised targets of embryonic 5mC promoter remodelling such as seven members of the *Pim* proto-oncogene, serine/threonine kinase-related family (pimr): *pimr57*, *pimr130*, *pimr109*, *pimr137*, *pimr128*, *pimr49* and *pimr51*, among others. To explore the developmental expression patterns of these 5mC-regulated genes, we plotted normalised RNA-seq counts across all examined stages for both PGC and soma samples (Fig. 5b). We found that his group of genes is developmentally downregulated in both PGCs and soma, albeit in PGCs their steady-state transcript levels are higher at 24−36 hpf, when compared to somatic cells. Notably, the major reduction in transcript levels for this gene group coincides with de novo deposition of promoter 5mC at 24 hpf (Fig. 5a). However, it is currently unclear what the exact role of 5mC is in this process. Both 5mC-regulated genes, such as *ddx4*, and genes whose promoters remain hypomethylated throughout embryogenesis (i.e. *cxcr4b*) (Fig. 5c), display similar developmental expression patterns, even though their abundances can vary greatly between PGCs and soma, as is the case for *ddx4* (Supplementary Fig. 8a, b).

In mammals, promoters of germline-expressed genes become methylated shortly after implantation, in the early post-implantation epiblast[96,97]. Given the strong germline enrichment of zebrafish embryonic 5mC promoter targets (Supplementary Dataset 15), we explored the degree of evolutionary conservation of this phenomenon. We identified orthologues of these putative 5mC-regulated promoters and plotted their 5mC levels in murine intracellular cell mass (ICM) sample, epiblasts (E6.5, E7.5) and male and female PGCs (E13.5)[16] (Fig. 5d). This analysis resulted in the identification of 23 high-confidence orthologues, 20 of which displayed increased 5mC during epiblast stages. As observed in their zebrafish counterparts, these promoters were specifically methylated in the oocytes but not in sperm. Furthermore, in line with global 5mC erasure during PGC migration[13], these targets were fully unmethylated in both male and female mouse PGCs at E13.5 (Fig. 5d, Supplementary Fig. 8c). Whereas some of the genes, such as *Sycp1*, *Sycp3* and *Dazl*, have previously been described as methylated in mouse embryos[96,97], this analysis also revealed additional 5mC-regulated promoters that are methylated in both zebrafish and mice and that could be of potential relevance for vertebrate embryogenesis. Those include *zar1l*, *tdrd9*, *tdrd12* and others (Fig. 5d). In summary, here we demonstrate shared 5mC targeting principles between the developing zebrafish germline and the soma and show that embryonic 5mC deposition on germline gene promoters is a conserved feature of vertebrate embryogenesis.

**Conserved 5mC germline targets are enriched in CTAs**. Because of the strong germline ontology enrichment and sperm-specific hypomethylation of the conserved 5mC promoter targets (Fig. 5a–d), we next wanted to explore whether this group of genes is enriched in cancer testis antigens (CTAs)[98]. In mammals,

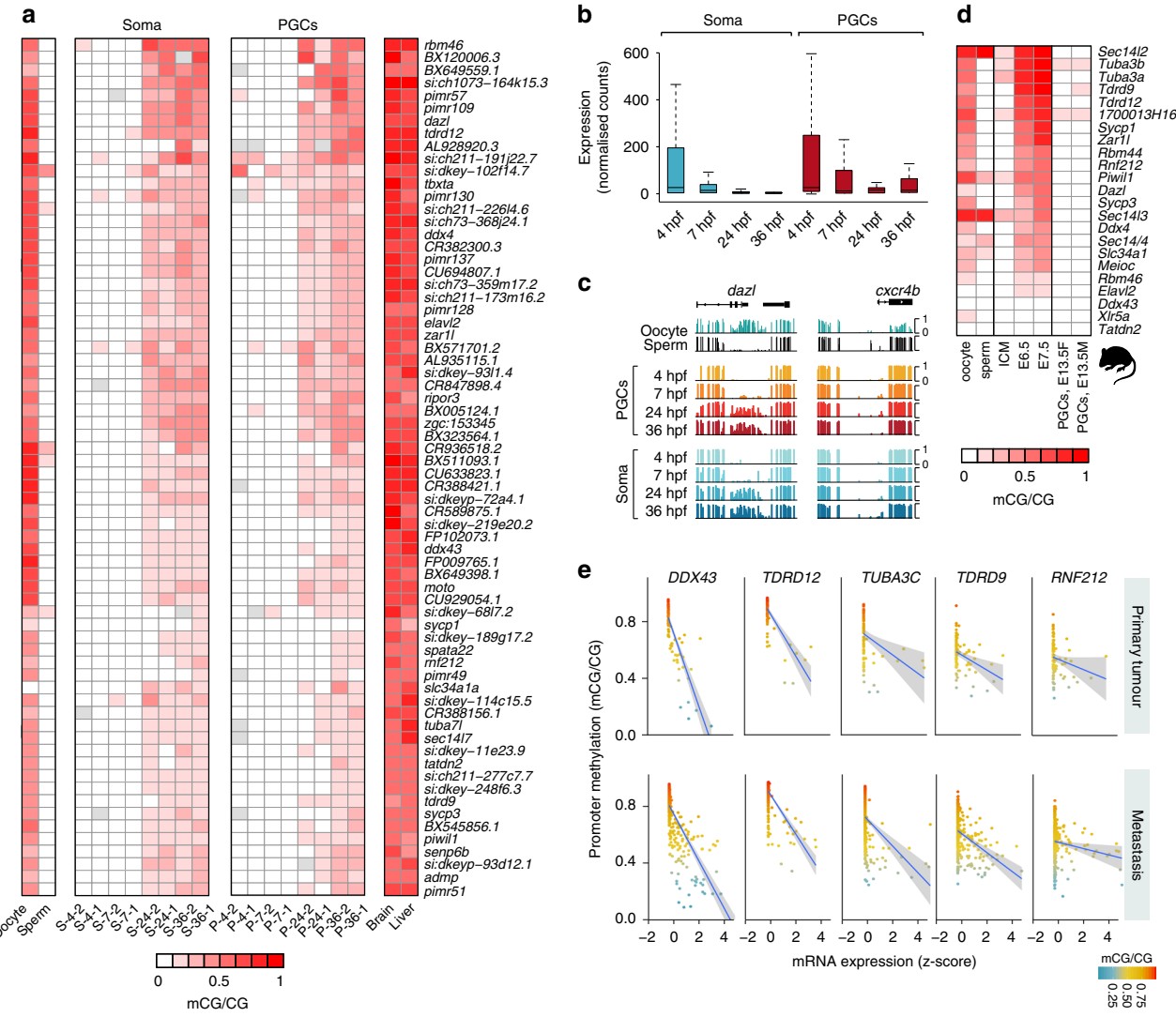

**Fig. 5** Embryonic 5-methylcytosine (5mC) targeting of germline gene promoters. **a** Heatmap of 5mC in promoter-associated CpG islands (Bio-CAP) targeted by 5mC during zebrafish embryogenesis. The status of gene promoter 5mC is shown in adult germ cells (oocyte, sperm), embryonic somatic cells and primordial germ cells (PGCs), as well as in adult somatic tissues (brain, liver). **b** Boxplots showing expression dynamics of genes whose promoters are targeted by 5mC during embryogenesis in PGCs and soma. The boxes show the interquartile range (IQR) around the median. The upper and lower whiskers extend from the hinge to the largest and smallest value, respectively, no further than 1.5 IQR. **c** Browser tracks showing developmental 5mC profiles at *dazl* and *cxcr4b* promoters. **d** Heatmap of 5mC at promoter-associated CpG islands (Bio-CAP) in mouse adult germ cells (oocyte, sperm), early embryos (intracellular cell mass (ICM), E6.5, E7.5) and developing PGCs. **e** Scatter plots of mean promoter 5mC values (methylation β values—450 K beadChIP arrays) and expression (*z*-scores, RNA-seq V2 RSEM) of the corresponding genes in The Cancer Genome Atlas-skin cutaneous melanoma (TCGA-SKCM) patient samples (primary tumour and metastasis). The lines represent a linear regression model, whereas the shaded areas represent the extent of standard error (SE)

CTAs are mostly expressed in the male germline, whereas their expression in somatic tissues is highly restricted. CTAs are frequently reactivated in human cancers, including melanoma and breast cancer, where they are believed to contribute to specific features of the neoplastic phenotype, including invasiveness and metastatic capacity[99]. Three of the conserved embryonic 5mC targets: *ddx4*, *sycp1*, and *rbm46*, are *bona fide* CTAs as per the CTA database[100]. To explore the links between 5mC-mediated germline targeting and oncogenesis more comprehensively, we identified human orthologs of zebrafish embryonic 5mC targets[86] and assessed their expression pattern in diverse human tissues[101]. Strikingly, these orthologues (*n* = 32) display strong testis-specific expression indicative of their potential CTA function (Supplementary Fig. 9a). To provide a proof of principle that these 5mC targets silenced during vertebrate embryogenesis are reactivated

in human cancers, and that this reactivation correlates with loss of promoter 5mC, we searched The Cancer Genome Atlas (TCGA) database[102] for skin cutaneous melanoma samples for which 5mC and RNA expression data were made available. We chose the melanoma due to frequent CTA reactivation in this type of cancers[99]. We observed considerable variability in 5mC patterns associated with CpG islands of eight conserved 5mC targets (Supplementary Fig. 9b). Comparisons with expression data revealed an anti-correlation between CpG island 5mC content and the expression of five of these targets in melanoma (Fig. 5e). This pattern was clearly visible for *DDX43*, *TDRD9*, *TDRD12*, *TUBA3C*, and *RNF212* (Ring finger protein 212 *RNF212*). While *DDX43*, *TDRD9*, *TDRD12* and *TUBA3C* display strict testis-specific expression patterns, *RNF212* was expressed more broadly, however with a notable enrichment in testis and

the ovary. RNF212 is a probable E3-SUMO protein ligase that plays major roles during meiotic crossover[103,104]. Such meiotic regulators when aberrantly expressed in mitotic cells could lead to failures associated with chromosome segregation and thus contribute to oncogenesis[105]. Altogether, these data demonstrate that evolutionarily conserved promoter 5mC targets, which become silenced during embryogenesis in both mammals and ana-mniotes, display predominantly male-germline specific expression patterns and are reactivated in human malignancies.

## Discussion

In this study we set out to describe the epigenome and the transcriptome of zebrafish PGCs covering four stages of embryonic development. We found that the zebrafish PGC DNA methylome is largely maintained in its paternal configuration during those embryonic stages. This is in line with previous studies that demonstrated the absence of genome-wide DNA methylome reprogramming in zebrafish and *Xenopus* embryos[21–24], and that described maternal to paternal 5mC reconfiguration in whole zebrafish embryos prior to ZGA[25,26]. This work thus provides the missing link in the understanding of how anamniote vertebrates program their developmental and germline epigenomes. Furthermore, these findings provide support to the idea that the two rounds of 5mC reprogramming observed in mammals, during preimplantation development and PGC formation, are likely linked to mammalian-specific processes, such as X-chromosome inactivation and imprint resetting[12,13]. In line with our observations, Ortega-Recalde et al. demonstrate the absence of genome wide 5mC reprogramming in 14 additional time-points spanning early PGC development (24–48 hpf), gonadal primordium (2–11 dpf), juvenile ovary (11–21 dpf), and early gonad transformation (25–28 dpf) stages[106]. Interestingly, they identify a period linked to zebrafish feminisation during which female-specific germline amplification and demethylation of an 11.5 kb repeat region encoding 45S ribosomal RNA (fem-rDNA) is taking place. This is in agreement with previous work that demonstrated how 5-aza-dC, a 5mC inhibitor, can induce long-term changes in the gonads and feminise zebrafish[107].

Based on the current data, we cannot completely exclude the possibility that more extensive DNA methylome reprogramming might occur before 4 hpf, or later during spermatogenesis and oogenesis stages. It is worth noting, however, that TET expression is extremely low in pre-gastrula embryos and that no hmC enrichment was observed by immunofluorescence in zebrafish 3 hpf (1K cell) to 10 hpf (tailbud) embryos[30]. While these results argue against the existence of active TET-mediated DNA demethylation in the developing zebrafish germline, the possibility of earlier TET-independent removal of DNA methylation as reported in mouse zygotes[17], cannot be completely excluded. Transgenic tools allowing for lifetime labelling of embryonic, juvenile and male and female germlines will be crucial to answering questions related to 5mC remodelling during the teleost life cycle[108]. Assuming that there is no DNA methylome reprogramming in the zebrafish germline, one might wonder whether more variability in 5mC patterning could be observed among individual fish. While such an experiment has not been performed to date, it is worth noting that zebrafish are highly polymorphic and that this likely has profound impacts on DNA methylome patterning[109], thus providing more 5mC variation within the zebrafish population. In fact, the zebrafish genome sequencing project identified ~7 million SNPs between just two homozygous zebrafish individuals, which is significantly more than what is observed in humans[62]. It is also worth noting that zebrafish are not prone to inbreeding[110], which could potentially be caused by the accumulation of epimutations that are not reset during germline reprogramming.

In the current study, we demonstrate that paternal epigenetic memory is preserved almost in its entirety during the first 7 h of zebrafish PGC development. At 24 hpf, demethylation of regulatory regions is occurring in both PGCs and in somatic cells, albeit with different dynamics (Fig. 3). In parallel, targeted methylation of promoters implicated in germline regulation, such as *ddx4*, *ddx43*, *dazl*, *tdrd9* and *tdrd12*, is taking place during that same period. These promoters are targeted by 5mC in both the germline and the soma and they are methylated in adult tissues such as brain and liver and in mature oocytes (Fig. 5a). This suggests that the development of female germline and somatic tissues might utilise similar regulatory pathways and that these genes might require silencing for the proper development of the presumptive ovary in juvenile fish. Notably, many of those transcripts are contributed to the embryo maternally[44], which is interesting from the point of view that their promoters are methylated in oocytes. This could be explained by a couple of scenarios. First, these promoters might be temporally unmethylated during later stages of oogenesis to allow for their expression, before 5mC-dependent silencing would occur in mature oocytes. Alternatively, it is possible that methylated promoters are expressed specifically during oocyte development. Whereas expression from methylated promoters is not common, strong evidence suggests that this phenomenon is more widespread than previously thought[23,24,111,112]. This is supported by recent studies, which demonstrated robust interaction of myriad transcription factors with methylated DNA sequences[113,114]. Furthermore, a group of methylated promoters was recently shown to be expressed during mammalian male germline development[111].

Our DMR analysis unravelled a small set of regulatory regions that display differential 5mC between PGCs and somatic cells. These DMRs are located in gene-distal regions and can be grouped into two major classes: DMRs hypomethylated in 4 hpf PGCs and DMRs hypermethylated in 24 hpf PGCs. The latter group is enriched in 5hmC in 24 hpf embryos, which is indicative of active DNA demethylation occurring at those regions. The temporal retention of 5(h)mC observed in PGCs but not in somatic cells at these loci at 24 hpf might be explained by differing cell cycle lengths or tissue-specific presence or absence of transcription factors implicated in 5mC removal. Overall, our data demonstrate the absence of major PGC-specific 5mC remodelling events during the first 36 h of zebrafish embryogenesis. This is in line with observations that the major source of difference in RNA and protein abundances between somatic and germline tissues is linked to the differential stabilisation of transcripts[84,115] and poly(A)-tail-independent non-canonical translation[116].

Finally, through evolutionary comparisons we demonstrated that despite highly diverse germline formation strategies, a number of PGC-enriched genes are expressed during zebrafish, mice and human germline development. We have expanded the catalogue of these evolutionarily conserved germline-expressed genes to include less well-characterised ones such as *golga3*, *gpat2*, *kpna6* and *adad1*. Future studies will explore their functional relevance within the context of vertebrate germline formation, to determine whether they are indeed required for the establishment or maintenance of PGC cell fate. Perhaps most interestingly, our study identified a set of promoters that become methylated during zebrafish and mammalian embryogenesis and that are commonly reactivated in human cancers such as melanoma. These genes display highly restricted patterns of somatic expression and are robustly expressed in the male germline. It remains to be determined whether this highly specific 5mC embryonic targeting has been conserved over such large evolutionary distances due to the deleterious effects that the somatic expression of these targets might have on cellular integrity. In

summary, our high-resolution DNA methylome and transcriptome maps of zebrafish PGC development reveal common and species-specific principles of germline reprogramming among vertebrates, and provide a useful resource for the studies of epigenetic regulation of germline cell fate.

## Methods

**Zebrafish housing and husbandry.** Transgenic zebrafish line used in this work is Tg(kop:egfp-f′ nos3′ UTR-cry:dsred), expressing farnesylated EGFP in PGCs[53]. Zebrafish were maintained on a 14-h light/10-h dark cycle, and fertilised eggs were collected and raised at 25.1, 27.7 or 32 °C. Embryos were kept in 0.3× Danieau's solution [17.4 mM NaCl, 0.21 mM KCl, 0.12 mM MgSO4·7H2O, 0.18 mM Ca (NO3)2, 1.5 mM HEPES (pH 7.6)]. To inhibit pigmentation in embryos older than 24 hpf, 0.003% N-phenylthiourea (Sigma) in 0.3× Danieau's solution was used. The study complies with all relevant ethical regulations for animal testing and research. The general fish maintenance at the Institute follows the regulations of the LANUV NRW and is supervised by the veterinarian office of the city of Muenster (Germany).

**Isolation of zebrafish PGCs.** PGCs and somatic cells were sorted from 4, 7, 24 and 36 hpf embryos carrying the transgene using FACS [FACSAria cell sorter (BD Biosciences) equipped with a 70-μm nozzle].

**Genomic DNA extraction from zebrafish PGCs and somatic cells.** Zebrafish PGCs and their corresponding somatic cells (n = 1000–14,000) were homogenised in 500 μL of homogenisation buffer (20 mM Tris, pH 8.0, 100 mM NaCl, 15 mM EDTA, 1% SDS, 0.5 mg/μL proteinase K). The homogenate was incubated for 3 h at 55 °C. The tubes were inverted 3–5 times during incubation to ensure optimal proteinase K cleavage. Two times phenol/chloroform/isoamylalcohol (25:24:1, PCI Sigma-Aldrich) extractions were performed using 500 μL of PCI. The samples were then spun for 10 min at 21,130 × g in a table-top centrifuge following each PCI extraction. The DNA was precipitated by adding 1/10 volume of 4 M NH4Ac, 1 volume of cold isopropanol (Sigma-Aldrich), and 20 μg/mL of linear acrylamide (Ambion). DNA was precipitated for 1 h on ice followed by centrifugation for 20 min at 4 °C and 21,130 × g in a table-top centrifuge. The DNA pellet was washed with 500 μL of 70% ethanol (Sigma Aldrich) and centrifuged for 5 min, 21,130 × g at room temperature. The pellet was resuspended in 200 μL of nuclease-free water (not DEPC-treated, Ambion) with RNAse A (20 μg/μL) and left at room temperature for 30 min. The RNAse A reaction was precipitated with 0.1 volumes of 4 M NH4Ac, 1 volume of isopropanol (Sigma-Aldrich) and 20 μg/mL of linear acrylamide (Ambion), on ice for 1 h. The precipitate was then centrifuged for 20 min at 4 °C at 21,130 × g in a table-top centrifuge and washed with 70% ethanol (Sigma-Aldrich). The pellet was resuspended in 30 μL of nuclease-free water (not DEPC-treated, Ambion).

**Low-input WGBS library preparation.** Genomic DNA extracted from zebrafish PGCs and their corresponding somatic cells (1–5 ng, 125 μL total volume) was spiked with unmethylated lambda phage fragments (Promega), sonicated on the M220 focused ultrasonicator (Covaris) with the following settings: peak incident power, 50 W; duty factor, 20%; cycles per burst, 200; treatment time, 75 s. Sonicated DNA was concentrated in a vacuum centrifuge concentrator to the final volume of 20 μL, required for bisulfite conversion with the EZ DNA Methylation-Gold Kit (Zymo Research). Bisulfite-converted DNA was then subjected to low input library preparation using the Accel-NGS Methyl-seq DNA Kit (Swift Biosciences). Briefly, the single-stranded, bisulfite-converted DNA was subjected to an adaptase reaction, followed by primer extension, adapter ligation (Methyl-seq Set A Indexing Kit— Swift Biosciences), and indexing PCR (13 cycles). Library size and consistency was determined by the Agilent 4200 Tapestation system. The libraries were quantified using the KAPA Library Quantification Kit (Roche) yielding ~10–20 nM.

**Zebrafish WGBS methylome data analysis.** Zebrafish PGC and soma DNA methylome libraries were sequenced on the Illumina HiSeqX platform (high-throughput mode, 150 bp, PE), generating an average of 75 M reads per sample. Sequenced reads in FASTQ format were trimmed using the fastp v0.12.5 tool (https://github.com/OpenGene/fastp) with the following settings: (fastp -i ${read_1} -I ${read_2} -o ${trimmed_read_1} -O ${trimmed_read_2} -f 10 -t 10 -F 10 -T 10). This step removes potential adaptor sequences and trims 10 bp of the 5′ and 3′ end of each read (required due to the addition of adaptase tails during library preparation). Trimmed reads were mapped (danRer10 genome reference, containing the lambda genome as chrLambda) using WALT[117] with the following settings: -m 10 -t 24 -N 10000000 -L 2000. The average mappability of the libraries was 43%. Mapped reads in SAM (Sequence Alignment/Map) format were converted to BAM ((Binary Alignment/Map)) and indexed using SAMtools[118]. PCR and optical duplicates were removed using Picard tools v2.3.0. Genotype and methylation bias correction was performed using MethylDackel software (https://github.com/dpryan79/MethylDackel). The number of methylated and unmethylated calls at each genomic CpG position were determined using MethylDackel

(MethylDackel extract genome_lambda.fa $input_bam -o output --mergeContext --minOppositeDepth 5 --maxVariantFrac 0.5 --OT 0,120,0,120 --OB 10,0,0,0). Only cytosines in the CpG dinucleotide context were retained for further analyses. Bisulfite conversion efficiency was estimated from the lambda phage spike-ins. To assess the 5mC abundance and dynamics at repetitive elements, sequencing reads were trimmed using fastp v0.12.5 and mapped using WALT v1.0, as described above, to an in silico reference consisting of danRer10 repeat-masked genome and single canonical repeat sequences from Repbase (https://www.girinst.org/repbase/). The numbers of methylated and unmethylated calls at each genomic CpG position were called using MethylDackel v0.3.0 with additional parameters: --keepSingleton, --keepDiscordant.

**WGBS methylome analysis.** PCA of PGCs, corresponding somatic cells and whole embryo methylomes (256-cell embryos SRP020008), as well as adult germline (oocyte (1), sperm (1) GSE44075; oocyte (2), sperm (2) SRP020008) and somatic methylomes (adult brain GSE68087; adult liver—this study) was performed using prcomp() R function and plotted using geom_point() function {ggplot2 v3.1.0}. Average 5mC values were calculated for 10 kbp non-overlapping bins across the genome for each sample using overlapRatios() function (https://github.com/astatham/aaRon/blob/master/R/overlaps.R).

**Comparative zebrafish and mammalian PGC methylome analysis.** To identify putative 5mC-regulated gene promoters, we first calculated average CpG methylation per promoter-associated non-methylated island, NMI (GSE43512)[75]. NMI peaks were called using MACS2 (narrowPeak)[119] separately for zebrafish testis, liver, and 24 hpf embryo samples, concatenated and merged. This NMI set was then overlapped with GRCz10/danRer10 Ensembl gene transcription start sites. Average CpG methylation was calculated for each NMI in soma samples across four developmental stages. NMIs with Pearson's correlation coefficients between 5mC and developmental stages >0.5 (indicative of a gradual increase of 5mC in somatic cells across stages) with 4 hpf 5mC <0.025 and 36 hpf 5mC >0.15 in each replicate and displaying >0.45 5mC in adult zebrafish brain and liver tissues were defined as putative 5mC targets. Statistical significance of this developmental 5mC increase was tested using Fisher's exact test (R 3.5.0) under the default parameters. To this end, fisher.test R function was applied to each NMI for the combined 5mC values from 4 hpf/7 hpf and 24 hpf/36 hpf embryonic data. Obtained p values were adjusted for multiple comparisons using Benjamini–Hochberg correction (p.adjust R fuction). NMIs with p.adj <0.01 in both replicates were marked as significant. To determine whether 5mC dynamics of putative CTA promoters is conserved in mammalian embryos, we utilised public bisulfite sequencing data from mouse sperm, oocyte, early embryo (ICM, E6.5, E7.5) and PGCs (E13.5 female and E13.5 male)[16] (GSE56697). Mouse orthologues were identified using ENSEMBL BioMart[86]. Transcription start sites of these orthologous genes were overlapped with NMIs (merged testis, liver and ESC BioCAP peaks[75] and average 5mC was calculated per NMI.

**Identification of DMRs.** Differentially methylated CpG sites between PGCs and soma at each developmental stage were identified using DMLtest function of the DSS package[120]. For 7, 24 and 36 hpf time points DMLtest was performed using biological replicates per each time point. For 4 hpf time point DMLtest was performed separately for each technical replicate with the commonly identified differentially methylated CpGs being used in the downstream analyses. Differentially methylated CpG sites located within 50 bp from each other when joined into regions (DMRs), which were filtered to harbour at least five differentially methylated CpGs and span at least 50 bp. DMRs overlapping repetitive regions (RepeatMasker) and/or containing more than 25% of CpGs within them with sequencing coverage <5× were excluded from the analysis. The remaining DMRs were defined as hypomethylated or hypermethylated when both replicates displayed at least 10% average methylation difference between PGCs and soma.

**Analysis of DNA methylation in clinical cancer samples.** To assess the 5mC status of gene promoters in clinical cancer cases, we utilised HumanMethylation 450K beadChIP (chromatin immunoprecipitation) arrays CpG methylation data from TCGA Project[102]. We focused on the skin cutaneous melanoma cohort composed of 89 primary tumour samples, 345 metastatic samples and 2 normal samples. Promoter 5mC was calculated as average 450K CpG probe 5mC across overlapping NMIs (merged BioCAP peaks from human testis and liver that overlapped hg19 Ensembl gene transcription start sites). A linear regression model (lm) generated in R (geom.smooth) was used to establish the correlation between 5mC and transcription.

**ChIP-seq data analysis.** H3K4me1 and H3K27ac ChIP-seq data from zebrafish dome and 24 hpf stage embryos[73] (GSE32483) and BioCAP data from zebrafish testis, liver and 24 hpf embryos[75] (GSE43512) was mapped to danRer10 zebrafish reference genome using bowtie v1.1.0 allowing up to three mismatches[121]. Reads mapping to multiple genomic locations or PCR duplicates[118] were discarded. BioCAP peak calling was performed using MACS2[119]. H3K4me1/H3K27ac enrichment over DMRs and size-distribution-matched controls was calculated using bamCoverage function (RPKM normalisation mode) from deepTools2[122].

BioCAP signal enrichment over DMRs relative to the input in the form of log 2 ratio was calculated using *bamCompare* function (RPKM normalisation mode) from deepTools2[122].

**RNA extraction from zebrafish PGCs and somatic cells**. Total RNA from germ cells and somatic cells was isolated using the PicoPure RNA Extraction Kit as per the manufacturer's instructions (Arcturus; Alphametrix).

**RNA-seq library preparation**. RNA-seq libraries were prepared with 1–3 ng input RNA material using SMARTer Stranded Total RNA-seq Kit v2—Pico Input Mammalian (Takara Bio USA Inc., USA), according to the manufacturer's instructions. RNA quality control was performed using Total RNA 6000 Pico Kit (RNA integrity number >8).

**RNA-seq data analysis**. Illumina Adapters and Pico v2 SMART adapter trimming (first three nucleotides of R2) was performed using TrimGalore v0.4.0 (https://github.com/FelixKrueger/TrimGalore). Trimmed sequencing reads were aligned to the zebrafish reference genome danRer10 using STAR v2.4.0d[123]. Quantification of transcript abundances was performed using RSEM v1.2.21[124]. Differential gene expression analysis was performed using edgeR[81]. Genes that displayed ±1.5 logFC (FDR <0.05) between PGCs and soma samples were considered as significantly differentially expressed.

**AS analysis**. To quantify AS, we used vast-tools v2 (https://github.com/vastgroup/vast-tools; species key "Dre")[94]. To compare different groups, we required that all eight samples had a sufficient read coverage (corresponding to a VLOW in vast-tools combine output). Then, we performed paired comparisons of exon inclusion levels using the percent-spliced-in (PSI) metric between tissues (PGC vs. soma at each developmental stage) or developmental stages (7 hpf vs. 24 hpf in PGC or soma). We required a minimum average delta PSI (dPSI) of 15 and of 5 for each of the individual paired comparisons for an exon to be considered as differentially regulated. Gene ontology analyses were performed using TopGO (http://bioconductor.org/packages/release/bioc/html/topGO.html), using as background all multiexonic genes harbouring at least one AS event with sufficient read coverage. Finally, to identify consistent tissue-specific exon markers, we selected those exons with (i) a minimum dPSI of 10 in each tissue comparison, (ii) an average dPSI higher than 15, and (iii) a p value lower than 0.05 in a t test comparing all four PGC and soma samples.

**Comparative zebrafish and mammalian PGC RNA-seq analysis**. For genes upregulated at minimum two consecutive developmental stages in zebrafish PGCs, mouse and human homologues were identified. Mouse single-cell RNA-seq data for E11.5, E13.5 female, E16.5 female and E16.5 male PGCs and corresponding soma samples were downloaded from GSE79552[40]. Human single-cell RNA-seq data for 4W, 7W, 8W, 11W male/female and 17W female and 19W male embryos containing the highest number of sequenced single PGCs and corresponding somatic cells were downloaded from GSE63818[41]. Mean FPKM values across single cells for each sample were calculated and plotted in a log-transformed format per each gene.

**Reporting summary**. Further information on research design is available in the Nature Research Reporting Summary linked to this article.

## Data availability
WGBS, MethylC-seq and RNA-seq data generated in this study are available from NCBI Gene Expression Omnibus[125] (GEO SuperSeries accession number GSE122723). Specifically, PGC and somatic cell WGBS data areavailable under GSE122722. Adult liver MethylC-seq data is available under GSE123493. PGC and somatic cells RNA-seq data is available under GSE122480. Additional RNA-seq data of 7 and 24 hpf PGC and somatic cells used for the alternative splicing analysis together with GSE122480 is available under GSE128986. All other relevant data supporting the key findings of this study are available within the article and its Supplementary Information files or from the corresponding author upon reasonable request. The Source Data underlying Figs. 1a, 2a–d, 3a–f, 4a–f, 5a–e and Supplementary Figs. 3a, 7d, 8a–d and 9a are provided as Source Data files 1–9, respectively. A reporting summary for this Article is available as a Supplementary Information files.

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

## Acknowledgements

We would like to thank Phuc-Loi Luu for assistance with whole-genome bisulfite sequencing analysis, and Alex de Mendoza and Devon Ryan for advice related to analyses of repetitive genomic sequences. We also thank Michael Geng for assistance with extraction of RNA. O.B. is supported by NHMRC (R.D. Wright Biomedical CDF APP1162993) and CINSW (CDF181229).

## Author contributions

O.B. and E.R. designed the study. K.S., K.T., and R.L. contributed to concept and study design. Zebrafish work was performed by K.T. and E.R. M.S. performed FACS sorting of zebrafish PGCs. Methyl-seq and RNA-seq library preparation and sequencing was carried out by K.S. K.S. analysed the data with input from O.B. AS analysis was performed by M.I. K.S and O.B. wrote the manuscript with input from all authors. All authors discussed the results and commented on the manuscript.

## Additional information

**Competing interests:** The authors declare no competing interests.

