## [Peer Review File · Nature Communications]

Reviewers' comments:

Reviewer #1 (Remarks to the Author):

The study by Skvortsova and colleagues provides the first high resolution analysis of DNA methylomes through progression of PGC development. Interestingly and in contrast to mammalian PGC development, DNA methylation is retained and specifically exhibits patterns similar to the sperm methylome. Accompanying transcriptomes from matched stages revealed the PGC marker genes were not subjected to dynamic changes to DNA methylation states. It was fascinating to learn that there are a set of genes that are dynamically modified by DNA methylation and that this is shared between somatic and PGC cells. This also coincided nicely with TET1 expression and the authors even included analysis of 5hmC at these sites to show its enrichment. These data collectively reveal that DNA demethylation occurs over time as a result of targeting by Tet. Many of the PGC specifically expressed genes are also conserved with mice and humans, indicating the quality of the isolated targeted cells is of good quality. Overall, I find this to be a very exciting and well-done study combining high resolution epigenomic analysis with the biology of a very interesting cell type. The comments below are intended to improve this manuscript.

Minor comments:

1. Figure 1c requires an x-axis label
2. Figure 1 – d and e are swapped in the figure labelling. In general, I don't see a need for 1e. The correlations could just be stated in the text.
3. The definition of CpG islands should be clearly explained given the distinction to the classic, yet arbitrary, definition used to define them in mammals. The current presentation is only meant for those who specialized in understanding non-mammalian vertebrate DNA methylation.
4. I find the use of the word "argue" overused and unnecessarily confrontational. It is used throughout this manuscript.
5. The section on alternative splicing seems out of place and really doesn't add much to the story. It could be removed or moved to the supplement.
6. The lines drawn on Figure 5b are odd. It's a forced line that doesn't represent the data well. Just present the dot plots as is without the line.

Reviewer #2 (Remarks to the Author):

In the manuscript, "Retention of paternal epigenetic memory in the developing teleost germline", Skvortsova et al report whole genome bisulfite sequencing of DNA extracted zebrafish embryos and primordial germ cells (PGCs) as four stages of development. Using this data, they address a longstanding question in the field of chromatin biology/germ cell development, which is whether or not the methylation state of non-mammalian vertebrate PGCs is reprogrammed in the same way as it is in mammals. Importantly, they find that, at least during the first 36 hours of development, the methylome of PGCs look like sperm, indicating paternal epigenetic memory. These observations are bolstered by transcriptome data identifying genes that are uniquely transcribed in PGCs and additional observations noting that many of the genes that are specifically targeted by DNA methylation in somatic and germline tissues during development are also misregulated in cancer.

Although the work is primarily descriptive in nature, the new data sets on methyl/transcriptomes of zebrafish PGCs are likely to be of high value to multiple communities, and the finding of paternal epigenetic memory provides an important new insight with relevance to those in the fields of germ cell development, DNA methylation, and reprogramming. Major and minor points of concern are listed below.

Major:

1. The authors use transcript levels for key germline markers to show that their PGC samples are enriched in this cell type. However, there isn't any data speaking to the fraction of somatic

contamination in PGC samples. Can the authors provide some data to address this? One approach could be to assess the fraction of cells that are GFP positive after sorting. Analysis of transcripts in RNA seq data that should be somatic only could also be useful.

2. The methods section indicates that Repeatmasker was used to mask repeats before 5mC analysis was performed. Is this true for figure 1C or only for DMR analysis? While there are certainly challenges associated with analysis of repeats, given that the bulk of 5mC resides in these sequences, it seems like more attention could be paid to their behavior in this manuscript. At the very least, the potential for unreported DMRS to reside in repeats should be addressed in the discussion.

3. I was somewhat confused by the section entitled "somatic and germline targets of promoter 5mC". It seems like the authors shift their DMR calling strategy in this section on a subset of promoters, and suggest that in contrast to previous analysis these sequences are in fact DMRs. Why do that just for these sequences? If this reduced stringency is also valid, why not apply it genome wide?

Minor:

For accuracy, the title should be either:
Retention of paternal epigenetic memory in a developing teleost germline
Or Retention of paternal epigenetic memory in the developing zebrafish germline.
One germline is not enough to definitively speak for all teleosts.

In figure 1 a, the authors refer to embryos as being injected with the kop-egfp transgene at 4 hpf ect. Surely, if they were injected it was at the one cell stage? This should be revised to be more clear, and if true details about amount of plasmid injected ect should be included in methods. However, this description seems to contradict the methods section, where the experiments are said to be carried out in embryos carrying the transgene, with reference to a stable transgenic line. Clarification is needed.

Figure 1 d and e legends appear transposed.

Figure 3a legend does not explain mir430 data clearly.

Figure 3 b Perhaps label samples in the figure as PGCs for clarity?

Supplemental 1. It would be easier to read these panels if the legend was presented in the same order as the data on panels (this would also place both the legend and panels in chronological order which would be easier to read).

Reviewer #3 (Remarks to the Author):

Skvortsova and colleagues characterize the DNA methylome in early zebrafish PGCs using WGBS. In the four developmental stages analysed, they do not find evidence of a PGC specific global DNA demethylation event – which has been described in mouse and human. In addition, they characterise gene-expression changes and find that germ line gene expression appears not to be regulated by DNA methylation. Finally, they identify genes that are targeted for DNA methylation during early embryogenesis and that are also dysregulated in human cancer.

Overall this is an interesting and well-conducted study, by researchers with significant expertise the field. This will undoubtedly be of general interest, particularly to those studying germ line development and epigenetic reprogramming. However, there are some issues that need to be addressed prior to publication.

1. Developmental stages analysed. There is insufficient discussion/explanation of the stages

analysed and the relatively limited timescale assessed impacts the conclusions that can be drawn. This requires significant attention before the paper is ready for publication. The authors do explain their selection strategy to include stages equivalent to the period of epigenetic reprogramming in mouse PGCs using RBTS. However, it is not necessarily the case that DNA demethylation occurs at the same development stage in zebrafish – especially given that there are significant differences in germ line development between mice and zebrafish. I have a number of related questions/concerns:

- is it possible to isolate PGCs any earlier than 3hpf? If so, is it possible that the authors missed an early period of DNA demethylation and subsequent remethylation?

- a figure showing how the timings selected map onto zebrafish PGC development would be helpful.

- Based on the data presented it seems possible that DNA demethylation might occur at a later stage? For instance, one another reason to suggest this is the pattern of DNA methylation at germ line genes which are methylated throughout the period studied and include relatively late markers in the mouse (including *ddx4* and *dazl*). As a rule such genes become expressed and and/or their regulatory elements become DNA demethylated around the time of global DNA demethylation in mice. When are these genes (and other examples such as *Nanos3/Dnd1*) demethylated in zebrafish PGCs? Can the authors rule out that this is the period which is accompanied by or correlated with global DNA demethylation?

- Page 6: 'could be potentially linked to sex-determination in juvenile fish' – the authors do not appear to present data to back up this statement?

In general, the authors should moderate their claims throughout the manuscript to make clear that they can only rule out DNA demethylation during the timepoints analysed. Alternatively, they may refer to the co-submitted manuscript by Hore and colleagues to support their claims. In addition, the term 'epigenetic reprogramming' should not be used interchangeably with DNA demethylation as it is possible there may be DNA methylation independent epigenetic reprogramming (such as changes in histone modifications, histone exchange etc).

2. The quantification of DNA methylation is based solely on WGBS analysis. The merits of this as a methodology to assess global DNA methylation levels is disputed. In particular, the presence of repetitive elements with unknown copy number is problematic – and they likely represent a significant proportion of global DNA methylation. Have the authors validated their findings with an orthogonal quantitative method, such as LC-MS?

3. If there is no reprogramming of DNA methylation, is there evidence of more variability in DNA methylation patterns between zebrafish individuals (as compared to mice)? Would the authors expect differences to emerge in different populations (or across generations), as any sporadic accumulation of DNA methylation would not be reset in the germ line? Might this impact the CG content in the genome over evolutionary time?

4. Introduction

- contrary to the authors' assertion, it has now been established that Tet enzymes are not required for active DNA demethylation in either the zygotic or PGCs {Amouroux: 2016be, Hill: 2018cq}.

- 'reacquisition of totipotency during pre-implantation development'. It is not clear what this means. Totipotency is present in the zygote, rather than acquired during pre-implantation development. Perhaps the authors are referring to the acquisition of pluripotency in the epiblast?

- 'lack the initial wave of global DNA demethylation' Does this refer to active DNA demethylation of the paternal genome? Or both this and the subsequent passive DNA demethylation that occurs thereafter.

- 'phenomenon specific to mammals'. What about plants?

- 'Our data demonstrate the absence of genome wide reprogramming events'. This statement is too strong and nonspecific. The authors focus on a short timescale and almost entirely on DNA methylation.

5. Cancer section: In my view, the study is sufficiently interesting without the human cancer section, which could be removed. I'm not sure this builds greatly on current knowledge regarding CTAs and reactivation of germ line markers in cancer or really develops the story in a meaningful way. In many ways, it's a distraction.

6. Discussion:

- Page 16: enhancer demethylation is not really depicted in Figure 2 (?Supp Figure 2C).
 - Page 16: 'Surprisingly' – is not a good choice, as this appears to be in keeping with mammals, and is rather expected.
 - Page 16: 'remain methylated... in mature oocytes'. This implies stability of the DNA methylome in oocytes – which the authors have not addressed. There could easily be dynamic changes.
 - Page 17. 'Our data argues against major PGC-specific 5mC remodelling' – far too strong a statement based on the data presented.
- Page 17 – hydroxymethylation. Is there sufficient enrichment to be confident that 5hmC is actually leading to significant DNA demethylation? (i.e. how do 5hmC levels equate to 5mC levels?).

7. Other issues:

- Figure 1. Egg samples. What stage of oogenesis is analysed? It seems surprising that the eggs cluster with somatic cells, as they have much lower levels of DNA methylation globally. Also, is there an explanation for the difference between the two replicates?
- Figure 1: legends d and e are swapped
- Page 9: The statement 'arguing against a major role for 5mC in PGC fate determination'. Is this really a suggested mechanism of fate determination in zebrafish PGCs? (it isn't thought to be in mice). How would this work in the context of germ plasm?
- Section 'Somatic and germline targets of promoter 5mC'. It appears that the promoters discussed behave differently to the global pattern, and actually increase DNA methylation to be more similar to the female pattern. This could perhaps be explained/discussed more clearly, as some readers may find this surprising/counterintuitive.

Reviewers' comments:

Reviewer #1 (Remarks to the Author):

The study by Skvortsova and colleagues provides the first high resolution analysis of DNA methylomes through progression of PGC development. Interestingly and in contrast to mammalian PGC development, DNA methylation is retained and specifically exhibits patterns similar to the sperm methylome. Accompanying transcriptomes from matched stages revealed the PGC marker genes were not subjected to dynamic changes to DNA methylation states. It was fascinating to learn that there are a set of genes that are dynamically modified by DNA methylation and that this is shared between somatic and PGC cells. This also coincided nicely with TET1 expression and the authors even included analysis of 5hmC at these sites to show its enrichment. These data collectively reveal that DNA demethylation occurs over time as a result of targeting by Tet. Many of the PGC specifically expressed genes are also conserved with mice and humans, indicating the quality of the isolated targeted cells is of good quality. Overall, I find this to be a very exciting and well-done study combining high resolution epigenomic analysis with the biology of a very interesting cell type. The comments below are intended to improve this manuscript.

Response: We thank the reviewer for the constructive comments and support of the manuscript.

Minor comments:

1. Figure 1c requires an x-axis label

Response: The x-axis label ("5mC levels (mCG/CG)") has now been added. Fig.1 is now **Fig. 2**

2. Figure 1 – d and e are swapped in the figure labelling. In general, I don't see a need for 1e. The correlations could just be stated in the text.

Response: We have corrected the labelling of Figure 2d/e. Fig. 1e is now **Supplementary Fig. 2c**. The correlations are now stated in the text.

3. The definition of CpG islands should be clearly explained given the distinction to the classic, yet arbitrary, definition used to define them in mammals. The current presentation is only meant for those who specialized in understanding non-mammalian vertebrate DNA methylation.

Response: We thank the Reviewer for this suggestion. A thorough explanation of CpG islands and how they are defined in this study has now been added to the Results section, where the CpG islands are first mentioned:

“Similarly, these DMRs overlapped with CpG islands, which are short genomic sequences of high CpG density that: i) are usually hypomethylated in vertebrate genomes, ii) that frequently coincide with gene-regulatory regions, and iii) that can be subject to 5mC-mediated regulation (Bird, 1986; Long, 2013). CpG islands were initially defined based solely on their sequence content (Gardiner-Garden, 1987; Takai, 2002), however such algorithms did not perform well on anamniote genomes (Han, 2008). We thus utilised genomic coordinates of CpG islands identified through the Bio-CAP (non-methylated DNA pulldown) approach throughout this study (Long, 2013) (**Fig. 3**).”

4. I find the use of the word “argue” overused and unnecessarily confrontational. It is used throughout this manuscript.

Response: We thank the Reviewer for this suggestion. The word “argue” has now been removed and those sentences modified.

5. The section on alternative splicing seems out of place and really doesn't add much to the story. It could be removed or moved to the supplement.

Response: We agree with the Reviewer that the section on alternative splicing (AS) did not add much to the story in its present form. The reason why we explored AS in the first place was due to the fact that 1) AS is known to contribute to cell fate specification, and 2) based on the current literature it appears that RNA-based mechanisms (i.e differential 3'UTR usage, miRNA targeting) are critical for PGC determination. We have now reanalysed our AS experiments and have thoroughly re-written this part to describe both stage- and cell type-specific AS events. We found that the majority of AS events are stage-, rather than cell-type specific, which we believe is significant enough to be included. Nevertheless, we managed to obtain seven high confidence AS markers that are differentially utilised between somatic cells and PGCs. We believe that these new additions further clarify the role of AS during zebrafish embryogenesis and germline development and merit the inclusion in the manuscript. The new data is now presented as **Supplementary Fig. 6**.

6. The lines drawn on Figure 5b are odd. It's a forced line that doesn't represent the data well. Just present the dot plots as is without the line.

Response: We thank the Reviewer for pointing this out and we apologise for not making this clear. The lines represent a simple linear regression model (lm) generated in R (geom.smooth) whereas the shaded areas represent the extent of the standard error. These are fairly straightforward procedures frequently used in the representation of genomic patterns. We are happy to include this information in the figure legend and in the Methods section so that the readers can obtain an immediate insight into what the lines represent. We would, however, prefer to include the lines as we believe that they are useful in highlighting the pattern of anti-correlation between promoter 5mC and transcription.

Reviewer #2 (Remarks to the Author):

In the manuscript, "Retention of paternal epigenetic memory in the developing teleost germline", Skvortsova et al report whole genome bisulfite sequencing of DNA extracted zebrafish embryos and primordial germ cells (PGCs) as four stages of development. Using this data, they address a longstanding question in the field of chromatin biology/germ cell development, which is whether or not the methylation state of non-mammalian vertebrate PGCs is reprogrammed in the same way as it is in mammals. Importantly, they find that, at least during the first 36 hours of development, the methylome of PGCs look like sperm, indicating paternal epigenetic memory. These observations are bolstered by transcriptome data identifying genes that are uniquely transcribed in PGCs and additional observations noting that many of the genes that are specifically targeted by DNA methylation in somatic and germline tissues during development are also misregulated in cancer.

Although the work is primarily descriptive in nature, the new data sets on methyl/transcriptomes of zebrafish PGCs are likely to be of high value to multiple communities, and the finding of paternal epigenetic memory provides an important new insight with relevance to those in the fields of germ cell development, DNA methylation, and reprogramming. Major and minor points of concern are listed below.

Response: We thank the reviewer for the constructive comments and support of the manuscript. We are happy to address all the issues raised by the Reviewer.

Major:

1. The authors use transcript levels for key germline markers to show that their PGC samples are enriched in this cell type. However, there isn't any data speaking to the fraction of somatic contamination in PGC samples. Can the authors provide some data to address this? One approach could be to assess the fraction of cells that are GFP positive after sorting. Analysis of transcripts in RNA seq data that should be somatic only could also be useful.

Response: We thank the Reviewer for this suggestion. To address this point in the most quantitative manner, we now performed FACS sorting of the *kop:egfp-f'-nos3'UTR-cry:dsred* line, followed by resorting of the GFP+ population. Briefly, during the first sort, we obtained a total of 1,667 cells that were: viable, single, and GFP-positive. These cells were then subjected to a second round of sorting. Out of 582 cells that were viable and single in the second round of sorting, 566 (97.3%) were GFP-positive. We thus estimate the somatic contamination to be below 3%. These data are now presented as **Supplementary Fig. 1** and a sentence related to the purity of the PGC fraction has been added to the main text:

“The purity of the sorted PGC cells was estimated at > 97% (**Supplementary Fig. 1**).”

2. The methods section indicates that Repeatmasker was used to mask repeats before 5mC analysis was performed. Is this true for figure 1C or only for DMR analysis? While there are certainly challenges associated with analysis of repeats, given that the bulk of 5mC resides in these sequences, it seems like more attention could be paid to their behavior in this manuscript. At the very least, the potential for unreported DMRS to reside in repeats should be addressed in the discussion.

Response: As indicated in the manuscript, Repeatmasker filtering was performed only for the purpose of a more stringent DMR identification. The overall 5mC levels (now **Fig. 2b**) were calculated with repeats included. We have now added a supplementary table with DMRs found within repetitive DNA (**Supplementary Table 2**). Also, to address a point from Reviewer 3, which also touches on this topic, we have now performed mapping to an *in silico* generated reference genome that contains canonical repeat sequences as per the Rebase repository (<https://www.girinst.org/server/RepBase/index.php>) (please see Methods for more details). The reason for this experiment was that while the zebrafish genome is extremely repeat-rich (> 50%), it is likely that the location and sequence of these repeats varies from fish to fish, and many of the reads mapping to these regions would be discarded due to multi-mapping. We have thus created a reference genome containing single canonical repeat sequences from Rebase as well as the non-repeat portion of the zebrafish genome and performed WGBS mapping as described in Methods. While this approach does not provide information on the genomic position of repeats, it could potentially provide an answer as to whether there is any global 5mC reprogramming happening within repeat DNA. However, we did not find any evidence for this in either of our two replicate WGBS experiments (**Supplementary Fig. 3**).

3. I was somewhat confused by the section entitled “somatic and germline targets of promoter 5mC”. It seems like the authors shift their DMR calling strategy in this section on a subset of promoters, and suggest that in contrast to previous analysis these sequences are in fact DMRs. Why do that just for these sequences? If this reduced stringency is also valid, why not apply it genome wide?

Response: We apologise for the confusion. Most DMR-finding algorithms are unbiased in the sense that they search for differentially methylated CpGs across the genome and then merge them into larger regions of defined statistical significance. This is the approach that we used for identifying PGC/soma DMRs, since we had no previous knowledge as to where they might be located (**Fig. 3**). However, as discussed in the text, we and others (Bogdanovic et al, 2016; Potok et al 2013) have previously observed that certain germline CpG island (non-methylated island) promoters are targeted by 5mC in early embryos. Since in this case we were aware of their genomic location, we utilised the available (Long et al, 2013) set of empirically defined CpG island promoters (> 20,000) and calculated 5mC for each of them while trying to identify the ones that gain DNA methylation developmentally (please see Methods for more details). In the previous version we have not provided any statistical significance details for this approach. To address this, we now provide overall 5mC levels for each of those CpG islands, as well as adjusted P values obtained from Fisher’s exact test (**Supplementary Table 14**). In summary, here we wanted to better define the set of promoters that gain 5mC in the soma (Bogdanovic et al, 2016; Potok et al 2013) and explore whether the same phenomenon can be observed in PGCs. In **Fig. 5a** we demonstrate that all the regions that gain promoter 5mC in the soma also gain 5mC in PGCs thus validating our initial DMR approach that did not find any differences in promoter 5mC between PGCs and somatic cells. We have now clarified this in the text.

“Importantly, all of these promoters also displayed a developmental increase in 5mC in PGCs (**Fig. 5a**), in line with the absence of soma- or PGC-specific developmental promoter DMRs (**Supplementary Tables 2, 3**).”

Minor:

For accuracy, the title should be either:

Retention of paternal epigenetic memory in a developing teleost germline

Or Retention of paternal epigenetic memory in the developing zebrafish germline. One germline is not enough to definitively speak for all teleosts.

Response: We thank the Reviewer for this suggestion. We have renamed the manuscript to: “*Retention of paternal epigenetic memory in a developing teleost germline*”.

In figure 1 a, the authors refer to embryos as being injected with the kop-egfp transgene at 4 hpf ect. Surely, if they were injected it was at the one cell stage? This should be revised to be more clear, and if true details about amount of plasmid injected ect should be included in methods. However, this description seems to contradict the methods section, where the experiments are said to be carried out in embryos carrying the transgene, with reference to a stable transgenic line. Clarification is needed.

Response: We are grateful to the Reviewer for spotting this and we apologise for the error. This sentence has now been removed. Indeed, all the data presented in this manuscript were collected from the stable transgenic line (*kop:egfp-f'-nos3'UTR-cry:dsred*)(Blaser et al, 2005), and as correctly described in the Methods section.

Figure 1 d and e legends appear transposed.

Response: This is now fixed. Figure 1 is now **Fig. 2**.

Figure 3a legend does not explain mir430 data clearly.

Response: We have added the description of miRNA binding sites enrichment in the legend of Figure 3, now **Fig. 4**

Figure 3 b Perhaps label samples in the figure as PGCs for clarity?

Response: Done

Supplemental 1. It would be easier to read these panels if the legend was presented in the same order as the data on panels (this would also place both the legend and panels in chronological order which would be easier to read).

Response: This has now been rectified according to the Reviewer's suggestion.

Reviewer #3 (Remarks to the Author):

Skvortsova and colleagues characterize the DNA methylome in early zebrafish PGCs using WGBS. In the four developmental stages analysed, they do not find evidence of a PGC specific global DNA demethylation event –

which has been described in mouse and human. In addition, they characterise gene-expression changes and find that germ line gene expression appears not to be regulated by DNA methylation. Finally, they identify genes that are targeted for DNA methylation during early embryogenesis and that are also dysregulated in human cancer.

Overall this is an interesting and well-conducted study, by researchers with significant expertise the field. This will undoubtedly be of general interest, particularly to those studying germ line development and epigenetic reprogramming. However, there are some issues that need to be addressed prior to publication.

Response: We thank the Reviewer for the support of our manuscript, as well as for excellent comments and suggestions and a very thorough review in general.

1. Developmental stages analysed. There is insufficient discussion/explanation of the stages analysed and the relatively limited timescale assessed impacts the conclusions that can be drawn. This requires significant attention before the paper is ready for publication. The authors do explain their selection strategy to include stages equivalent to the period of epigenetic reprogramming in mouse PGCs using RBTS. However, it is not necessarily the case that DNA demethylation occurs at the same development stage in zebrafish – especially given that there are significant differences in germ line development between mice and zebrafish. I have a number of related questions/concerns:

- is it possible to isolate PGCs any earlier than 3hpf? If so, is it possible that the authors missed an early period of DNA demethylation and subsequent remethylation?

Response: This is an excellent suggestion and is something that we would very much like to explore in the future. Our current line (*kop:egfp-f'-nos3'UTR-cry:dsred*) only allows for the interrogation of the 4 - 36hpf window. There are certainly other lines that could serve for the purpose of exploring earlier PGC time points, however, obtaining these lines and breeding them to numbers sufficient for early PGC cell isolation and optimising the sorting parameters could easily take up to a year, as PGCs only correspond to a very small fraction of the overall cell population. We agree with the Reviewer that it is possible that we might have missed a window of early 5mC reprogramming. However, it is worth noting that no hmC enrichment in PGCs or (somatic cells) was observed in 1K cell embryos (3hpf) (Almeida et al, 2012). Furthermore, no TET expression can be detected in zebrafish embryos before late gastrula (8h) stages. We have now added this to the discussion:

“Based on the current data, we cannot completely exclude the possibility that more extensive DNA methylome reprogramming might occur before 4hpf, or

later during spermatogenesis and oogenesis stages. It is worth noting however, that TET expression is extremely low in pre-gastrula embryos and that no hmC enrichment was observed by immunofluorescence in zebrafish 3hpf (1K cell) to 10hpf (tailbud) embryos (Almeida, 2012).”

- a figure showing how the timings selected map onto zebrafish PGC development would be helpful.

Response: This is an excellent suggestion. We have now included such a schematic as **Fig. 1b**. We have also further clarified our choice of the examined developmental stages:

“The embryonic stages were chosen according to Reciprocal Best Transcriptome Similarity (RBTS) index (Irie, 2011), to match the developmental period of mouse PGC specification and DNA methylome reprogramming (Seisenberger, 2012; Hill, 2018) (**Fig. 1b**). Specifically, we wanted to capture the developmental period, which in mouse would correspond to the initial specification of PGCs and early demethylation (E6.25 - E8.5/E9.5), migration and colonisation of the genital ridge (E8.5/E9.5 - E10.5), and global DNA demethylation (E10.5 - E12.5/E13.5) (Hill, 2018). Furthermore, it is worth noting that while significant differences in germline development strategies exist between zebrafish and mammals, in both organisms this period is characterised by PGC migration (Eddy, 1975; Raz, 2003; Ohinata, 2009).”

Based on the data presented it seems possible that DNA demethylation might occur at a later stage? For instance, one another reason to suggest this is the pattern of DNA methylation at germ line genes which are methylated throughout the period studied and include relatively late markers in the mouse (including *ddx4* and *dazl*). As a rule such genes become expressed and and/or their regulatory elements become DNA demethylated around the time of global DNA demethylation in mice. When are these genes (and other examples such as *Nanos3/Dnd1*) demethylated in zebrafish PGCs? Can the authors rule out that this is the period which is accompanied by or correlated with global DNA demethylation?

Response: We thank the Reviewer for pointing this out. In the combined 17 time points from both manuscripts, no global DNA demethylation was observed during early PGC development (4hpf - 36hpf), gonadal primordium (2 - 11 dpf), ‘juvenile ovary’ (11-21 dpf) and early gonad transformation (25 - 28 dpf) stages. Nevertheless, we observe that germline genes including *ddx4/vasa/dazl* are targeted specifically by mC between 7hpf and 24hpf. More interestingly perhaps, the majority of CpG-rich germline gene promoters such as *ddx4/dazl* are unmethylated in the sperm while methylated in the oocytes (even though the *ddx4* germ-plasm RNA/proteins are abundant in oocytes).

Notably, sperm displays much higher mC levels when compared to oocytes (**Fig. 2d**). We thus believe that such questions could only be answered with the help of transgenic lines that would allow for lifetime labelling of the germline and that would cover the period of dimorphic gonad formation as well as early and late oogenesis and spermatogenesis. These possibilities have now been described at multiple places in the Discussion:

“In line with our observations, Ortega-Recalde et al (this issue) demonstrate the absence of genome wide 5mC reprogramming in 14 additional time-points spanning early PGC development (24hpf - 48hpf), gonadal primordium (2 – 11 dpf), juvenile ovary (11-21 dpf), and early gonad transformation (25 – 28 dpf) stages.”

“Based on the current data, we cannot completely exclude the possibility that more extensive DNA methylome reprogramming might occur before 4hpf, or later during spermatogenesis and oogenesis stages.”

“Transgenic tools allowing for lifetime labelling of embryonic, juvenile, and male and female germlines will be crucial to answering questions related to 5mC remodelling during the teleost life cycle (Ye, 2019).”

- Page 6: ‘could be potentially linked to sex-determination in juvenile fish’ – the authors do not appear to present data to back up this statement?

Response: We have now removed this sentence and cited the co-submitted manuscript in the Discussion.

“In line with our observations, Ortega-Recalde et al (this issue) demonstrate the absence of genome wide 5mC reprogramming in 14 additional time-points spanning early PGC development (24hpf - 48hpf), gonadal primordium (2 – 11 dpf), juvenile ovary (11-21 dpf), and early gonad transformation (25 – 28 dpf) stages. Interestingly, they identify a period linked to zebrafish feminisation during which female-specific germline amplification and demethylation of an 11.5-kb repeat region encoding 45S ribosomal RNA (fem-rDNA) is taking place. This is in agreement with previous work that demonstrated how 5-aza-dC, a 5mC inhibitor, can induce long-term changes in the gonads and feminize zebrafish (Ribas, 2017).”

In general, the authors should moderate their claims throughout the manuscript to make clear that they can only rule out DNA demethylation during the timepoints analysed. Alternatively, they may refer to the co-submitted manuscript by Hore and colleagues to support their claims. In addition, the term ‘epigenetic reprogramming’ should not be used interchangeably with DNA demethylation as it is possible there may be DNA methylation independent epigenetic reprogramming (such as changes in histone modifications, histone exchange etc).

Response: We have cited the co-submitted manuscript and adjusted our claims so that it is clear that they refer to the 4hpf - 36hpf time-window. We have replaced the term epigenome/epigenetic reprogramming with 5mC reprogramming in parts where we refer to our data or to 5mC reprogramming in general.

2. The quantification of DNA methylation is based solely on WGBS analysis. The merits of this as a methodology to assess global DNA methylation levels is disputed. In particular, the presence of repetitive elements with unknown copy number is problematic – and they likely represent a significant proportion of global DNA methylation. Have the authors validated their findings with an orthogonal quantitative method, such as LC-MS?

Response: We agree with the Reviewer that standard WGBS mapping pipelines might not be best suited for the measurement of repeat 5mC content, which forms a significant part (~50%) of the zebrafish genome. Indeed, repeats exist in many copies (that often display some form of sequence diversity), and standard WGBS mapping pipelines routinely remove reads that map to multiple positions in the genome thereby hindering accurate repeat 5mC measurements. To overcome both the issue of sequence diversity and multi-mapping reads, we generated a reference genome that consists of 2322 canonical repeat sequences as per the Repbase repository (<https://www.girinst.org/server/RepBase/index.php>), as well as the non-repetitive portion of the zebrafish genome. In this reference, each canonical repeat sequence is represented only once thereby allowing reads that would normally multi-map, to map to a single position of this *in silico* generated reference. Also, given that Repbase utilises consensus repeat sequences, it is likely that this would further increase the mappability of reads originating from repetitive DNA. Nevertheless, even with this approach we were not able to detect any differences in the overall 5mC repeat content (**Supplementary Fig. 3**). We thus believe that at this point there is not need to further investigate global 5mC of 4-36h PGCs. We would however be very interested to utilise LC-MS should we encounter a stage of germline development where more extensive 5mC remodelling is happening and where such a technology would help us to further dissect potential 5mC/hmC dynamics, as recently described (Hill, 2018).

3. If there is no reprogramming of DNA methylation, is there evidence of more variability in DNA methylation patterns between zebrafish individuals (as compared to mice)? Would the authors expect differences to emerge in different populations (or across generations), as any sporadic accumulation of DNA methylation would not be reset in the germ line? Might this impact the CG content in the genome over evolutionary time?

Response: This is an excellent question and is definitely something that deserves to be mentioned in the discussion. To our knowledge such an experiment has not been performed yet and most of the available zebrafish WGBS methylomes are generated from pooled embryo preparations. It is known, however, that zebrafish are highly polymorphic, and that this will undoubtedly have effects on methylome patterning of individual fish. The zebrafish genome sequencing project identified 7 million SNPs between two homozygous zebrafish individuals. This is more than what can be observed between any two humans and is nearly one-fifth of all SNPs measured among 1,092 human diploid genomes. It is also well-known that zebrafish are not prone to inbreeding, and perhaps the accumulation of epimutations could partly be responsible for that as well. Regarding the increase in CG content, we are not sure whether the lack of germline reprogramming would have a major impact on that sequence feature. Zebrafish genomes are heavily methylated (just like mammalian genomes) so there would be little opportunity for accumulation of additional 5mC, except for in CpG island promoters and enhancers that in most cases remain unmethylated throughout the life cycle. We have now included a paragraph in the discussion that deals with this topic:

“Assuming that there is no DNA methylome reprogramming in the zebrafish germline, one might wonder whether more variability in 5mC patterning could be observed among individual fish. While such an experiment has not been performed to date, it is worth noting that zebrafish are highly polymorphic and that this likely has profound impacts on DNA methylome patterning (Fraser, 2012), thus providing more 5mC variation within the zebrafish population. In fact, the zebrafish genome sequencing project identified ~7 million SNPs between just two homozygous zebrafish individuals, which is significantly more than what is observed in humans (Howe, 2013). It is also worth noting that zebrafish are not prone to inbreeding (Mrakovcic, 1979), which could potentially be caused by the accumulation of epimutations that are not reset during germline reprogramming.”

4. Introduction

- contrary to the authors' assertion, it has now been established that Tet enzymes are not required for active DNA demethylation in either the zygotic or PGCs {Amouroux:2016be, Hill:2018cq}.

Response: We have now amended our introduction to better reflect these findings.

“While the exact mechanism by which DNA demethylation occurs in the mammalian zygote remains a topic of debate (Gu, 2011; Wossidlo, 2011; Wang, 2014; Amouroux, 2016), recent data suggest that TET proteins are not directly implicated in the initial wave of paternal DNA demethylation, which due to its dynamics can also not be fully explained by passive 5mC dilution (Amouroux, 2016). Similarly, during mammalian PGC DNA methylome

reprogramming, TET proteins are not required for the initiation, but rather for the maintenance of global PGC demethylation (Hill, 2018).”

- ‘reacquisition of totipotency during pre-implantation development’. It is not clear what this means. Totipotency is present in the zygote, rather than acquired during pre-implantation development. Perhaps the authors are referring to the acquisition of pluripotency in the epiblast?

Response: We have now removed this sentence.

- ‘lack the initial wave of global DNA demethylation’ Does this refer to active DNA demethylation of the paternal genome? Or both this and the subsequent passive DNA demethylation that occurs thereafter.

Response: We thank the Reviewer for pointing this out. We have now clarified this:

“Zebrafish and other non-mammalian (anamniote) vertebrates, lack global 5mC erasure (Veenstra, 2001; Macleod, 1999; Bogdanovic, 2011; Hontelez, 2015; Jiang, 2013; Potok, 2013), which in mammals occurs after fertilisation and persists during blastula stages (Oswald, 2000; Mayer, 2000; Smith, 2012).

- ‘phenomenon specific to mammals’. What about plants?

Response: We have now removed this claim

- ‘Our data demonstrate the absence of genome wide reprogramming events’. This statement is too strong and nonspecific. The authors focus on a short timescale and almost entirely on DNA methylation.

Response: We have now edited this and other claims where it was not clear that we are only discussing the first 36h of zebrafish embryogenesis

5. Cancer section: In my view, the study is sufficiently interesting without the human cancer section, which could be removed. I’m not sure this builds greatly on current knowledge regarding CTAs and reactivation of germ line markers in cancer or really develops the story in a meaningful way. In many ways, it’s a distraction.

Response: We agree with the Reviewer that perhaps this section does not build greatly on the current knowledge regarding CTAs, and this was not our initial intention. What we demonstrate in this section and what was not clearly demonstrated before is that evolutionarily conserved (fish/mouse/human) 5mC targets are enriched in germline/CTA promoters. However, when we searched the existing literature for evidence of cancer-specific demethylation and transcriptional activation of these promoters, we could not find a study that comprehensively covered our CTAs of interest in a single cohort. We thus undertook the analyses based on TCGA data to comprehensively show in a single figure that evolutionarily conserved CTAs can be reactivated in cancer and that this coincides with their promoter demethylation. To make this message clearer, we have now renamed that section to: “Conserved 5mC germline targets are enriched in CTAs”, which we believe better fits the content of the presented data. We would, however, appreciate if the Reviewer would let us keep this section in the manuscript, as we feel that it provides an interesting link between highly specific and conserved embryonic 5mC targets and their demethylation in cancer.

6. Discussion:

Page 16: enhancer demethylation is not really depicted in Figure 2 (?Supp Figure 2C).

Response: We have now rephrased this to:

“At 24 hpf, demethylation of regulatory regions is occurring in both PGCs and somatic cells, albeit with different dynamics.”

- Page 16: ‘Surprisingly’ – is not a good choice, as this appears to be in keeping with mammals, and is rather expected.

Response: We have removed the word “Surprisingly”.

- Page 16: ‘remain methylated... in mature oocytes’. This implies stability of the DNA methylome in oocytes – which the authors have not addressed. There could easily be dynamic changes.

Response: We have replaced “remain” with “are”.

- Page 17. ‘Our data argues against major PGC-specific 5mC remodelling’ – far too strong a statement based on the data presented.

Response: We have now rephrased this:

“Our data thus demonstrate the absence of major PGC-specific 5mC remodelling events during the first 36 hours of zebrafish embryogenesis and suggest that 5mC likely only plays an auxiliary role during zebrafish germline development.”

Page 17 – hydroxymethylation. Is there sufficient enrichment to be confident that 5hmC is actually leading to significant DNA demethylation? (i.e. how do 5hmC levels equate to 5mC levels?).

Response: The hmC data are from our previous manuscript (Bogdanovic et al, 2016). In zebrafish, hmC was particularly abundant at conserved enhancers and was often detected at levels of 0.1-0.3 (hmCG/CG). Furthermore in that study we demonstrated that the abundance of hmC in 24hpf zebrafish embryos anti-correlates with the abundance of 5mC at the subsequent stage (48hpf), ie the more hmC we detected at 24hpf, the less 5mC we observed at 48hpf at those positions. The hmC (TAB-seq) and mC (WGBS) libraries in those experiments were made from the same DNA. Furthermore, triple tet (tet1/2/3) morpholine knockdown resulted in a significant increase of 5mC specifically over those hmC-enriched regions. We thus concluded that in zebrafish, at least during 24-48h stages, hmC is associated with active demethylation (Bogdanovic 2016).

7. Other issues:

- Figure 1. Egg samples. What stage of oogenesis is analysed? It seems surprising that the eggs cluster with somatic cells, as they have much lower levels of DNA methylation globally. Also, is there an explanation for the difference between the two replicates?

Response: The oocyte samples come from two different studies (Potok et al, 2013 Cell; Jiang et al, 2013 Cell) and correspond to mature unfertilised oocytes (squeezed from anesthetized females). Following the Reviewer's question we decided to re-analyse these datasets and determine whether global 5mC levels could be responsible for such a clustering pattern. Firstly, we utilised a different PCA algorithm (please see Methods for details) due to an issue raised online associated with the algorithm that we were using (from the Deeptools suite), <https://www.biostars.org/p/261940/>. We essentially see the same clustering pattern (only now separated on PC1, as expected, **Fig. 2c**). Indeed, we observe that the clustering pattern is likely driven by global 5mC levels that are higher in the sperm/early embryos and lower in somatic tissues and oocytes. We have now included this data as **Fig. 2d**. We have also added a sentence on previously discussed similarities between oocyte and adult somatic 5mC patterns in zebrafish:

“It is worth noting, however, that zebrafish oocytes also resemble adult somatic tissues in hypermethylation of germline and developmental promoters, and hypomethylation of housekeeping and terminal differentiation promoters (Potok, 2013).”

As for the difference in oocyte methylomes, these data are from previously published datasets (Potok, Jiang, 2013) and have been generated by different labs and different library prep strategies, which could to some extent influence the methylome patterns. The polymorphic nature of the zebrafish population could also play a role.

- Figure 1: legends d and e are swapped

Response: This is now corrected in the new **Fig. 2**.

- Page 9: The statement ‘arguing against a major role for 5mC in PGC fate determination’. Is this really a suggested mechanism of fate determination in zebrafish PGCs? (it isn’t thought to be in mice). How would this work in the context of germ plasm?

Response: We have now rephrased this to:

“Genomic intersections of the identified DMRs (**Fig. 3a**), with the regulatory domains (McLean, 2010) of putative PGC regulators (**Fig. 4**) revealed no overlap (data not shown), suggestive of 5mC not playing a major role in PGC marker regulation.

Section ‘Somatic and germline targets of promoter 5mC’. It appears that the promoters discussed behave differently to the global pattern, and actually increase DNA methylation to be more similar to the female pattern. This could perhaps be explained/discussed more clearly, as some readers may find this surprising/counterintuitive.

Response: We have now added a sentence to further clarify this:

“Notably, almost all of the identified 5mC target promoters were unmethylated in adult sperm while being methylated in the oocytes, which is in contrast to the predominantly paternal DNA methylome patterns observed during embryogenesis.”

REVIEWERS' COMMENTS:

Reviewer #1 (Remarks to the Author):

The authors have satisfactorily addressed my concerns. I would also like to note as an expert with WGBS data generation/analysis that the data can be used to estimate genome-wide levels of DNA methylation by calculating the percent of cytosines in raw sequenced reads across a range of samples. This can be used to estimate DNA methylation levels in any species regardless of a publicly available reference genome. The authors used an equally effective approach in their response to reviewer 2 and 3 that is more complicated than I think is necessary. Regardless, it is a suitable method that reveals their original conclusions are correct.

Reviewer #2 (Remarks to the Author):

All my concerns regarding the manuscript by Skvortsova et al have been addressed. This is a nice story.

Reviewer #3 (Remarks to the Author):

The authors produced an excellent rebuttal and the manuscript is much improved. I only have a few minor remaining comments.

1. 'The active mechanism entails enzymatic oxidation of genomic 5mC by TET (Ten-eleven-translocation) family enzymes' – perhaps better described as 'an active mechanism'. As the authors point out, this is not the active mechanism in the zygote or in PGCs.
2. '5mC levels reach their lowest point in the blastocyst followed by cell type-specific remethylation during gastrulation'. The lowest point of 5mC is in PGCs following epigenetic reprogramming.
3. 'germline is specified during gastrulation'. In mice, it is prior to gastrulation. In humans, we don't know.
4. 'likely employ more extensive 5mC remodelling only during gonadogenesis.' – do the authors mean gametogenesis?
5. 'germline-expressed genes are specifically methylated in the early preimplantation embryo, between the blastocyst and the epiblast stage'. This statement is incorrect. The promoters become methylated shortly after implantation, in the early post-implantation epiblast. The term 'epiblast stage' is not routinely used in mouse embryology (this would be confusing anyway - as it wouldn't be clear whether the authors are referring to the pre-implantation epiblast (in the blastocyst) or the post-implantation epiblast).
6. 'It is worth noting however, that TET expression is extremely low in pre-gastrula...'.- It should be mentioned that any early DNA demethylation could also occur in a TET-independent manner – as occurs in the mouse germline and zygote.
7. 'and suggest that 5mC likely only plays an auxiliary role during germline development' – I am not convinced this statement is backed up by the data shown, and is rather unnecessary. I would delete.

Reviewers' comments:

Reviewer #1 (Remarks to the Author):

The authors have satisfactorily addressed my concerns. I would also like to note as an expert with WGBS data generation/analysis that the data can be used to estimate genome-wide levels of DNA methylation by calculating the percent of cytosines in raw sequenced reads across a range of samples. This can be used to estimate DNA methylation levels in any species regardless of a publicly available reference genome. The authors used a equally effective approach in their response to reviewer 2 and 3 that is more complicated than I think is necessary. Regardless, it is a suitable method that reveals their original conclusions are correct.

Response: We thank the reviewer for the constructive comments and support of the manuscript.

Reviewer #2 (Remarks to the Author):

All my concerns regarding the manuscript by Skvortsova et al have been addressed. This is a nice story.

Response: We thank the reviewer for supporting the manuscript.

Reviewer #3 (Remarks to the Author):

The authors produced an excellent rebuttal and the manuscript is much improved. I only have a few minor remaining comments.

Response: We thank the Reviewer for the support of our manuscript and valuable comments.

1. 'The active mechanism entails enzymatic oxidation of genomic 5mC by TET (Ten-eleven-translocation) family enzymes' – perhaps better described as 'an active mechanism'. As the authors point out, this is not the active mechanism in the zygote or in PGCs.

Response: We have now replaced "The active mechanism" with "An active mechanism".

2. '5mC levels reach their lowest point in the blastocyst followed by cell type-specific remethylation during gastrulation'. The lowest point of 5mC is in PGCs following epigenetic reprogramming.

Response: We thank the Reviewer for pointing this out. We intended to say "the lowest point in the preimplantation embryo", and we appreciate that the current wording was causing confusion. We changed the sentence to now read as: "DNA demethylation takes place up until the blastocyst stage followed by cell type-specific remethylation during gastrulation."

3. 'germline is specified during gastrulation'. In mice, it is prior to gastrulation. In humans, we don't know.

Response: We thank the Reviewer for pointing this out. We changed the statement to read as: "In the former group the germline is specified in the early embryo by means of induction".

4. 'likely employ more extensive 5mC remodelling only during gonadogenesis.' – do the authors mean gametogenesis?

Response: We have now changed the statement to read as "likely employ more extensive 5mC remodelling only during later stages of gametogenesis."

5. 'germline-expressed genes are specifically methylated in the early preimplantation embryo, between the blastocyst and the epiblast stage'. This statement is incorrect. The promoters become methylated shortly after implantation, in the early post-implantation epiblast. The term 'epiblast stage' is not routinely used in mouse embryology (this would be confusing anyway - as it it wouldn't be clear whether the authors are referring to the pre-implantation epiblast (in the blastocyst) or the post-implantation epiblast).

Response: We thank the Reviewer for pointing this out. We have replaced the statement to now read as "In mammals, promoters of germline-expressed genes become methylated shortly after implantation, in the early post-implantation epiblast".

6. 'It is worth noting however, that TET expression is extremely low in pre-gastrula...'.- It should be mentioned that any early DNA demethylation could also occur in a TET-independent manner – as occurs in the mouse germline and zygote.

Response: We have added the sentence discussing the potential for TET-independent DNA demethylation in developing zebrafish germline prior to 4 hpf. It now reads: "Based on the current data, we cannot completely exclude the possibility that more extensive DNA methylome reprogramming might occur before 4 hpf, or later during spermatogenesis and oogenesis stages. It is worth noting however, that TET expression is extremely low in pre-gastrula embryos and that no hmC enrichment was observed by immunofluorescence in zebrafish 3 hpf (1K cell) to 10 hpf (tailbud) embryos ³⁰. While these results argue against the existence of active TET-mediated DNA demethylation in the developing zebrafish germline, one cannot exclude the possibility of TET-independent removal of DNA methylation as reported in mouse zygotes ¹⁷."

7. 'and suggest that 5mC likely only plays an auxiliary role during germline development' – I am not convinced this statement is backed up by the data shown, and is rather unnecessary. I would delete.

Response: We have now deleted that statement to read as: "Overall, our data thus demonstrate the absence of major PGC-specific 5mC remodelling events during the first 36 hours of zebrafish embryogenesis. This is in line with observations that the major source of difference in RNA and protein abundances between somatic and germline tissues is linked to the differential stabilisation of transcripts ^{84, 115} and poly(A)-tail-independent non-canonical translation (PAINT) ¹¹⁶."